# Gradient Descent Temporal Difference-difference Learning

## Abstract

Though widely useful in reinforcement learning, "semi-gradient" methods—including TD($\lambda$) and Q-learning—do not converge as robustly as gradient-based methods. Even in the case of linear function approximation, convergence cannot be guaranteed for these methods when they are used with off-policy training, in which an agent uses a behavior policy that differs from the target policy in order to gain experience for learning. To address this, alternative algorithms that are provably convergent in such cases have been introduced, the most well known being gradient descent temporal difference (GTD) learning. This algorithm and others like it, however, tend to converge much more slowly than conventional temporal difference learning. In this paper we propose *gradient descent temporal difference-difference (Gradient-DD) learning* in order to improve GTD2, a GTD algorithm (Sutton et al., 2009b), by introducing second-order differences in successive parameter updates. We investigate this algorithm in the framework of linear value function approximation, theoretically proving its convergence by applying the theory of stochastic approximation. Studying the model empirically on the random walk task, the Boyan-chain task, and the Baird's off-policy counterexample, we find substantial improvement over GTD2 and, in several cases, better performance even than conventional TD learning.

## 1 Introduction

Many of the recent practical successes of reinforcement learning have been achieved using "semi-gradient" methods—including TD($\lambda$) and Q-learning—in which bootstrapping is used to quickly estimate a value function. However, because they introduce bias by learning a bootstrapped estimate of the target rather than the target itself, semi-gradient methods do not converge as robustly as gradient-based methods (Sutton & Barto, 2018). Even in the relatively simple case of linear function approximation, convergence cannot be guaranteed for these methods when they are used with off-policy training (Baird, 1995), in which an agent uses a behavior policy that differs from the target policy in order to gain experience for learning.

To address this shortcoming and to ground value prediction in the framework of stochastic gradient descent, the gradient-based temporal difference algorithms GTD and GTD2 were introduced (Sutton et al., 2009a;b). These algorithms are compatible with both linear function approximation and off-policy training, ensuring stability with computational complexity scaling linearly with the size of the function approximator. Despite this theoretical assurance, empirical evidence suggests that their convergence is notably slower than conventional temporal difference (TD) learning, limiting their practical utility (Ghiassian et al., 2020; White & White, 2016). Building on this work, extensions to the GTD family of algorithms (see (Ghiassian et al., 2018) for a review) have allowed for incorporating eligibility traces (Maei & Sutton, 2010; Geist & Scherrer, 2014), non-linear function approximation such as with a neural network (Maei, 2011), and reformulation of the optimization as a saddle point problem (Liu et al., 2015; Du et al., 2017). However, due to their slow convergence, none of these stable off-policy methods are commonly used in practice.

In this work, we introduce a new gradient descent algorithm for temporal difference learning with linear value function approximation. This algorithm, which we call *gradient descent temporal difference-difference* (Gradient-DD) learning, is an acceleration technique that employs second-order differences in successive parameter updates. The basic idea of Gradient-DD is to modify the error objective function by additionally considering the prediction error obtained in the last time step, then to derive a gradient-descent algorithm

based on this modified objective function. In addition to exploiting the Bellman equation to get the solution, this modified error objective function avoids drastic changes in the value function estimate by encouraging local search around the current estimate. Algorithmically, the Gradient-DD approach only adds an additional term to the update rule of the GTD2 method, and the extra computational cost is negligible. We prove its convergence by applying the theory of stochastic approximation. This result is supported by numerical experiments, which also show that Gradient-DD obtains better convergence in many cases than conventional TD learning.

## 1.1 Related Work

In related approaches to ours, some previous studies have attempted to improve Gradient-TD algorithms by adding regularization terms to the objective function. These approaches have used $l_1$ regularization on weights to learn sparse representations of value functions Liu et al. (2012), or $l_2$ regularization on weights Ghiassian et al. (2020). Our work is different from these approaches in two ways. First, whereas these previous studies investigated a variant of TD learning with gradient corrections, we take the GTD2 algorithm as our starting point. Second, unlike these previous approaches, our approach modifies the error objective function by using a distance constraint rather than a penalty on weights. The distance constraint works by restricting the search to some region around the evaluation obtained in the most recent time step. With this modification, our method provides a learning rule that contains second-order differences in successive parameter updates.

Our approach is similar to trust region policy optimization (Schulman et al., 2015) or relative entropy policy search (Peters et al., 2010), which penalize large changes being learned in policy learning. In these methods, constrained optimization is used to update the policy by considering the constraint on some measure between the new policy and the old policy. Here, however, our aim is to find the optimal value function, and the regularization term uses the previous value function estimate to avoid drastic changes in the updating process.

Our approach bears similarity to the natural gradient approach widely used in reinforcement learning (Amari, 1998; Bhatnagar et al., 2009; Degris et al., 2012; Dabney & Thomas, 2014; Thomas et al., 2016), which uses the metric tensor to correct for the local geometry of the parameter space, and also features a constrained optimization form. However, Gradient-DD is distinct from the natural gradient. The essential difference is that, unlike the natural gradient, Gradient-DD is a trust region method, which defines the trust region according to the difference between the current value and the value obtained from the previous step. From the computational cost viewpoint, unlike natural TD (Dabney & Thomas, 2014), which needs to update an estimate of the metric tensor, the computational cost of Gradient-DD is essentially the same as that of GTD2.

## 2 Gradient descent method for temporal difference learning

### 2.1 Problem definition and background

In this section, we formalize the problem of learning the value function for a given policy under the Markov decision process (MDP) framework. In this framework, the agent interacts with the environment over a sequence of discrete time steps, $t = 1, 2, \ldots$. At each time step the agent observes a state $s_t \in \mathcal{S}$ and selects an action $a_t \in \mathcal{A}$. In response, the environment emits a reward $r_t \in \mathbb{R}$ and transitions the agent to its next state $s_{t+1} \in \mathcal{S}$. The state and action sets are finite. State transitions are stochastic and dependent on the immediately preceding state and action. Rewards are stochastic and dependent on the preceding state and action, as well as on the next state. The process generating the agent's actions is termed the behavior policy. In off-policy learning, this behavior policy is in general different from the target policy $\pi : \mathcal{S} \to \mathcal{A}$. The objective is to learn an approximation to the state-value function under the target policy in a particular environment:

$$V(s) = \mathrm{E}_\pi \left[ \sum_{t=1}^{\infty} \gamma^{t-1} r_t | s_1 = s \right], \tag{1}$$

where $\gamma \in [0, 1)$ is the discount rate.

In problems for which the state space is large, it is practical to approximate the value function. In this paper we consider linear function approximation, where states are mapped to feature vectors with fewer components

than the number of states. Specifically, for each state $s \in \mathcal{S}$ there is a corresponding feature vector $\mathbf{x}(s) \in \mathbb{R}^p$, with $p \leq |\mathcal{S}|$, such that the approximate value function is given by

$$V_{\mathbf{w}}(s) := \mathbf{w}^\top \mathbf{x}(s). \tag{2}$$

The goal is then to learn the parameters $\mathbf{w}$ such that $V_{\mathbf{w}}(s) \approx V(s)$.

## 2.2 Gradient temporal difference learning

A major breakthrough for the study of the convergence properties of MDP systems came with the introduction of the GTD and GTD2 learning algorithms (Sutton et al., 2009a;b). We begin by briefly recapitulating the GTD algorithms, which we will then extend in the following sections. To begin, we introduce the Bellman operator $\mathbf{B}$ such that the true value function $\mathbf{V} \in \mathbb{R}^{|\mathcal{S}|}$ satisfies the Bellman equation:

$$\mathbf{V} = \mathbf{R} + \gamma \mathbf{P} \mathbf{V} =: \mathbf{B} \mathbf{V},$$

where $\mathbf{R}$ is the reward vector with components $\mathrm{E}(r_n|s_n = s)$, and $\mathbf{P}$ is a matrix of the state transition probabilities under the target policy. In temporal difference methods, an appropriate objective function should minimize the difference between the approximate value function and the solution to the Bellman equation.

Having defined the Bellman operator, we next introduce the projection operator $\mathbf{\Pi}$, which takes any value function $\mathbf{V}$ and projects it to the nearest value function within the space of approximate value functions of the form Eqn. (2). Letting $\mathbf{X}$ be the matrix whose rows are $\mathbf{x}(s)$, the approximate value function can be expressed as $\mathbf{V}_{\mathbf{w}} = \mathbf{X}\mathbf{w}$. The projection operator is then given by

$$\mathbf{\Pi} = \mathbf{X}(\mathbf{X}^\top \mathbf{D} \mathbf{X})^{-1}\mathbf{X}^\top \mathbf{D},$$

where the matrix $\mathbf{D}$ is diagonal, with each diagonal element $d_s$ corresponding to the probability of visiting state $s$ under the behavior policy. We consider a general setting as in Sutton et al. (2009b;a), where the first state of each transition is chosen i.i.d. according to an arbitrary distribution that may be unrelated to $\mathbf{P}$. This setting defines a probability over independent triples of state, next state, and reward random variables, denoted $(s_n, s_{n+1}, r_n)$, with associated feature-vector random variables $\mathbf{x}_n = \mathbf{x}_{s_n}$ and $\mathbf{x}_{n+1} = \mathbf{x}_{s_{n+1}}$.

The natural measure of how closely the approximation $\mathbf{V}_{\mathbf{w}}$ satisfies the Bellman equation is the mean-squared Bellman error:

$$\mathrm{MSBE}(\mathbf{w}) = \|\mathbf{V}_{\mathbf{w}} - \mathbf{B}\mathbf{V}_{\mathbf{w}}\|_{\mathbf{D}}^2, \tag{3}$$

where the norm is weighted by $\mathbf{D}$, such that $\|\mathbf{V}\|_{\mathbf{D}}^2 = \mathbf{V}^\top \mathbf{D} \mathbf{V}$. However, because the Bellman operator follows the underlying state dynamics of the Markov chain, irrespective of the structure of the linear function approximator, $\mathbf{B}\mathbf{V}_{\mathbf{w}}$ will typically not be representable as $\mathbf{V}_{\mathbf{w}}$ for any $\mathbf{w}$. An alternative objective function, therefore, is the mean squared *projected* Bellman error (MSPBE), which is defined by Sutton et al. (2009b) as

$$J(\mathbf{w}) = \|\mathbf{V}_{\mathbf{w}} - \mathbf{\Pi}\mathbf{B}\mathbf{V}_{\mathbf{w}}\|_{\mathbf{D}}^2. \tag{4}$$

Following Sutton et al. (2009b), our objective is to minimize this error measure. As usual in stochastic gradient descent, the weights at each time step are then updated by $\Delta\mathbf{w} = -\alpha \nabla J(\mathbf{w})$, where $\alpha > 0$, and

$$-\frac{1}{2}\nabla J(\mathbf{w}_n) = -\mathrm{E}[(\gamma\mathbf{x}_{n+1} - \mathbf{x}_n)\mathbf{x}_n^\top][\mathrm{E}(\mathbf{x}_n\mathbf{x}_n^\top)]^{-1}\mathrm{E}(\delta_n\mathbf{x}_n). \tag{5}$$

We have also introduced the temporal difference error $\delta_n = r_n + (\gamma\mathbf{x}_{n+1} - \mathbf{x}_n)^\top \mathbf{w}_n$. Let $\boldsymbol{\eta}_n$ denote the estimate of $[\mathrm{E}(\mathbf{x}_n\mathbf{x}_n^\top)]^{-1}\mathrm{E}(\delta_n\mathbf{x}_n)$ at the time step $n$. Because the factors in Eqn. (5) can be directly sampled, the resulting updates in each step are

$$\delta_n = r_n + (\gamma\mathbf{x}_{n+1} - \mathbf{x}_n)^\top \mathbf{w}_n$$
$$\boldsymbol{\eta}_{n+1} = \boldsymbol{\eta}_n + \beta_n(\delta_n - \mathbf{x}_n^\top\boldsymbol{\eta}_n)\mathbf{x}_n$$
$$\mathbf{w}_{n+1} = \mathbf{w}_n - \alpha_n(\gamma\mathbf{x}_{n+1} - \mathbf{x}_n)(\mathbf{x}_n^\top\boldsymbol{\eta}_n). \tag{6}$$

These updates define the GTD2 learning algorithm, which we will build upon in the following section.

While the algorithm described above would be appropriate for on-policy learning, we are interested in the case of off-policy learning, in which actions are selected based on a behavior policy different from the target policy. If value estimation consists of the estimation of the expected returns, this off-policy setting involves estimating an expectation conditioned on one distribution with samples collected under another. GTD2 can be extended to make off-policy updates by using importance sampling ratios $\rho_n = \pi(a_n|s_n)/b(a_n|s_n) \geq 0$ where $a_n$ denotes the action taken at state $s_n$. The resulting modifications to the equations for updating $\boldsymbol{\eta}_n$ and $\mathbf{w}_n$ are as follows:

$$\begin{aligned} \boldsymbol{\eta}_{n+1} =& \boldsymbol{\eta}_n + \beta_n(\rho_n\delta_n - \mathbf{x}_n^\top\boldsymbol{\eta}_n)\mathbf{x}_n \\ \mathbf{w}_{n+1} =& \mathbf{w}_n - \alpha_n\rho_n(\gamma\mathbf{x}_{n+1} - \mathbf{x}_n)(\mathbf{x}_n^\top\boldsymbol{\eta}_n). \end{aligned} \tag{7}$$

## 3    Gradient descent temporal difference-difference learning

In this section we modify the objective function by additionally considering the difference between $\mathbf{V_w}$ and $\mathbf{V_{w_{n-1}}}$, which denotes the value function estimate at step $n-1$ of the optimization. We propose a new objective $J_{\text{GDD}}(\mathbf{w}|\mathbf{w}_{n-1})$, where the notation "$\mathbf{w}|\mathbf{w}_{n-1}$" in the parentheses means that the objective is defined given $\mathbf{V_{w_{n-1}}}$ of the previous time step $n-1$. Specifically, we modify Eqn. (4) as follows:

$$J_{\text{GDD}}(\mathbf{w}|\mathbf{w}_{n-1}) = J(\mathbf{w}) + \kappa\|\mathbf{V_w} - \mathbf{V_{w_{n-1}}}\|_{\mathbf{D}}^2, \tag{8}$$

where $\kappa \geq 0$ is a parameter of the regularization, and we assume that $\kappa$ is constant. We show in Section A.1 of the appendix that minimizing Eqn. (8) is equivalent to the following optimization

$$\arg\min_{\mathbf{w}} J(\mathbf{w}) \text{ s.t. } \|\mathbf{V_w} - \mathbf{V_{w_{n-1}}}\|_{\mathbf{D}}^2 \leq \mu \tag{9}$$

where $\mu > 0$ is a parameter which becomes large when $\kappa$ is small, so that the MSPBE objective is recovered as $\mu \to \infty$, equivalent to $\kappa \to 0$ in Eqn. (8).

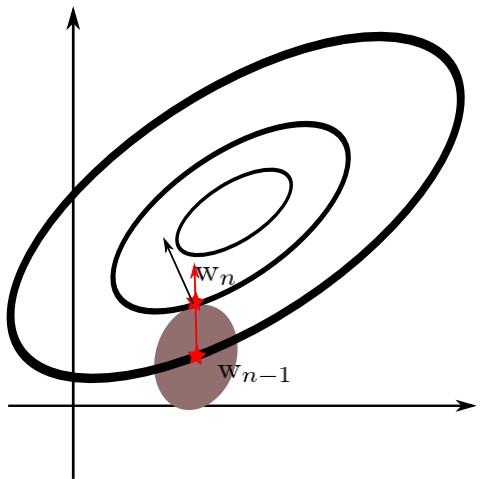

Figure 1: Schematic diagram of Gradient-DD learning with $\mathbf{w} \in \mathbb{R}^2$. Rather than updating $\mathbf{w}$ directly along the gradient of the MSPBE (black arrow), the update rule selects the direction starting from $\mathbf{w}_n$ (red star) that minimizes the MSPBE while satisfying the constraint $\|\mathbf{V_w} - \mathbf{V_{w_{n-1}}}\|_{\mathbf{D}}^2 \leq \mu$ (shaded ellipse).

Rather than simply minimizing the optimal prediction from the projected Bellman equation, the agent makes use of the most recent update to look for the solution, choosing a $\mathbf{w}$ that minimizes the MSPBE while following the constraint that the estimated value function should not change too greatly, as illustrated in Fig. 1. In effect, the regularization term encourages searching around the estimate at previous time step, especially when the state space is large.

Eqn. (9) shows that the regularized objective is a *trust region* approach, which seeks a direction that attains the best improvement possible subject to the distance constraint. The trust region is defined by the value

distance rather than the weight distance, meaning that Gradient-DD also makes use of the natural gradient of the objective around $\mathbf{w}_{n-1}$ rather than around $\mathbf{w}_n$ (see Section A.2 of the appendix for details). In this sense, our approach can be explained as a trust region method that makes use of natural gradient information to prevent the estimated value function from changing too drastically.

For comparison with related approaches using natural gradients, in Fig. 9 of the appendix we compare the empirical performance of our algorithm with natural GTD2 and natural TDC Dabney & Thomas (2014) using the random walk task introduced below in Section 5. In addition, we compared our approach of regularizing the objective using the difference in successive value estimates ($\kappa\|\mathbf{V}_{\mathbf{w}_n} - \mathbf{V}_{\mathbf{w}_{n-1}}\|_{\mathbf{D}}^2$) vs. using the difference in successive parameters ($\kappa\|\mathbf{w}_n - \mathbf{w}_{n-1}\|^2$). We found that, unlike Gradient-DD, the latter approach does not yield an improvement compared to GTD2 (Fig. 11 of the appendix).

With these considerations in mind, the negative gradient of $J_{\mathrm{GDD}}(\mathbf{w}|\mathbf{w}_{n-1})$ is

$$
\begin{aligned}
&-\frac{1}{2}\nabla J_{\mathrm{GDD}}(\mathbf{w}_n|\mathbf{w}_{n-1}) \\
&= -\mathrm{E}[(\gamma\mathbf{x}_{n+1} - \mathbf{x}_n)\mathbf{x}_n^\top][\mathrm{E}(\mathbf{x}_n\mathbf{x}_n^\top)]^{-1}\mathrm{E}(\delta_n\mathbf{x}_n) - \kappa\mathrm{E}[(\mathbf{x}_n^\top\mathbf{w}_n - \mathbf{x}_n^\top\mathbf{w}_{n-1})\mathbf{x}_n].
\end{aligned}
\tag{10}
$$

Because the terms in Eqn. (10) can be directly sampled, the stochastic gradient descent updates are given by

$$
\begin{aligned}
\delta_n &= r_n + (\gamma\mathbf{x}_{n+1} - \mathbf{x}_n)^\top\mathbf{w}_n \\
\boldsymbol{\eta}_{n+1} &= \boldsymbol{\eta}_n + \alpha_n(\delta_n - \mathbf{x}_n^\top\boldsymbol{\eta}_n)\mathbf{x}_n \\
\mathbf{w}_{n+1} &= \mathbf{w}_n - \kappa\alpha_n(\mathbf{x}_n^\top\mathbf{w}_n - \mathbf{x}_n^\top\mathbf{w}_{n-1})\mathbf{x}_n - \alpha_n(\gamma\mathbf{x}_{n+1} - \mathbf{x}_n)(\mathbf{x}_n^\top\boldsymbol{\eta}_n).
\end{aligned}
\tag{11}
$$

Similar to the case of GTD2 for off-policy learning in (7), the modifications to the equations for updating $\boldsymbol{\eta}_n$ and $\mathbf{w}_n$ for off-policy learning with Gradient-DD are as follows:

$$
\begin{aligned}
\boldsymbol{\eta}_{n+1} &= \boldsymbol{\eta}_n + \alpha_n(\rho_n\delta_n - \mathbf{x}_n^\top\boldsymbol{\eta}_n)\mathbf{x}_n \\
\mathbf{w}_{n+1} &= \mathbf{w}_n - \kappa\alpha_n\rho_n(\mathbf{x}_n^\top\mathbf{w}_n - \mathbf{x}_n^\top\mathbf{w}_{n-1})\mathbf{x}_n - \alpha_n\rho_n(\gamma\mathbf{x}_{n+1} - \mathbf{x}_n)(\mathbf{x}_n^\top\boldsymbol{\eta}_n).
\end{aligned}
\tag{12}
$$

These update equations define the Gradient-DD method, in which the GTD2 update equations (6) are generalized by including a second-order update term in the third update equation, where this term originates from the squared bias term in the objective (8). Since Gradient-DD is not sensitive to the step size of updating $\boldsymbol{\eta}$ (see Fig. 8 in the appendix), the updates of Gradient-DD only have a single shared step size $\alpha_n$ rather than two step sizes $\alpha_n, \beta_n$ as GTD2 and TDC used. It is worth noting that the computational cost of our algorithm is essentially the same as that of GTD2. In the following sections, we shall analytically and numerically investigate the convergence and performance of Gradient-DD learning.

## 4 Convergence Analysis

In this section we establish that the asymptotic convergence guarantees of the original GTD methods also apply to the Gradient-DD algorithm. Denote $\mathbf{G}_n = \begin{bmatrix} -\mathbf{x}_n\mathbf{x}_n^\top & -\mathbf{x}_n(\mathbf{x}_n - \gamma\mathbf{x}_{n+1})^\top \\ (\mathbf{x}_n - \gamma\mathbf{x}_{n+1})\mathbf{x}_n^\top & \mathbf{0} \end{bmatrix}$, and $\mathbf{H}_n = \begin{bmatrix} \mathbf{0} & \mathbf{0} \\ \mathbf{0} & \mathbf{x}_n\mathbf{x}_n^\top \end{bmatrix}$. We rewrite the update rules in Eqn. (11) as a single iteration in a combined parameter vector with $2n$ components, $\boldsymbol{\rho}_n = (\boldsymbol{\eta}_n^\top, \mathbf{w}_n^\top)^\top$, and a new reward-related vector with $2n$ components, $\mathbf{g}_{n+1} = (r_n\mathbf{x}_n^\top, \mathbf{0}^\top)^\top$, as follows:

$$
\boldsymbol{\rho}_{n+1} = \boldsymbol{\rho}_n - \kappa\alpha_n\mathbf{H}_n(\boldsymbol{\rho}_n - \boldsymbol{\rho}_{n-1}) + \alpha_n(\mathbf{G}_n\boldsymbol{\rho}_n + \mathbf{g}_{n+1}),
\tag{13}
$$

**Theorem 1.** *Consider the update rules (13) with step-size sequences $\alpha_n$. Let the TD fixed point be $\mathbf{w}^*$, such that $\mathbf{V}_{\mathbf{w}^*} = \mathbf{\Pi}\mathbf{B}\mathbf{V}_{\mathbf{w}^*}$. Suppose that (A0) $\alpha_n \in (0,1)$, $\sum_{n=1}^\infty \alpha_n = \infty$, $\sum_{n=1}^\infty \alpha_n^2 < \infty$, (A1) $(\mathbf{x}_n, r_n, \mathbf{x}_{n+1})$ is an i.i.d. sequence with uniformly bounded second moments, (A2) $E[(\mathbf{x}_n - \gamma\mathbf{x}_{n+1})\mathbf{x}_n^\top]$ and $E(\mathbf{x}_n\mathbf{x}_n^\top)$ are non-singular, (A3) $\sup_n \|\boldsymbol{\rho}_{n+1} - \boldsymbol{\rho}_n\|$ is bounded in probability, (A4) $\kappa$ is a constant such that $0 \le \kappa < \infty$. Then as $n \to \infty$, $\mathbf{w}_n \to \mathbf{w}^*$ with probability 1.*

*Proof sketch.* Due to the second-order difference term in Eqn. (13), the analysis framework in (Borkar & Meyn, 2000) does not directly apply to the Gradient-DD algorithm when (A0) holdes, i.e., step size is tapered. Likewise, the two-timescale convergence analysis (Bhatnagar et al., 2009) is also not directly applicable. Defining $\mathbf{u}_{n+1} = \boldsymbol{\rho}_{n+1} - \boldsymbol{\rho}_n$, we rewrite the iterative process in Eqn. (13) into two parallel processes which are given by

$$\boldsymbol{\rho}_{n+1} = \boldsymbol{\rho}_n - \kappa\alpha_n\mathbf{H}_n\mathbf{u}_n + \alpha_n(\mathbf{G}_n\boldsymbol{\rho}_n + \mathbf{g}_{n+1}), \tag{14}$$

$$\mathbf{u}_{n+1} = -\kappa\alpha_n\mathbf{H}_n\mathbf{u}_n + \alpha_n(\mathbf{G}_n\boldsymbol{\rho}_n + \mathbf{g}_{n+1}). \tag{15}$$

We analyze the parallel processes Eqns. (14) & Eqn. (15) instead of directly analyzing Eqn. (13). Our proofs have three steps. First we show $\sup_n \|\boldsymbol{\rho}_n\|$ is bounded by applying the stability of the stochastic approximation (Borkar & Meyn, 2000) into the recursion Eqn. (14). Second, based on this result, we shall show that $\mathbf{u}_n$ goes to 0 in probability by analyzing the recursion Eqn. (15). At last, along with the result that $\mathbf{u}_n$ goes to 0 in probability, by applying the convergence of the stochastic approximation (Borkar & Meyn, 2000) into the recursion Eqn. (14), we show that $\boldsymbol{\rho}_n$ goes to the TD fixed point which is given by the solution of $\mathbf{G}\boldsymbol{\rho} + \mathbf{g} = 0$. The details are provided in Section A.3 of the Appendix. □

Theorem 1 shows that Gradient-DD maintains convergence as GTD2 under some mild conditions. The assumptions (A0), (A1), and (A2) are standard conditions in the convergence analysis of Gradient TD learning algorithms (Sutton et al., 2009a;b; Maei, 2011). The assumption (A3) is weak since it means only that the incremental update in each step is bounded in probability. The assumption (A4) requires that $\kappa$ is a constant, meaning $\kappa = O(1)$. Given this assumption, the contribution of the term $\kappa\mathbf{H}_n\mathbf{u}_n$ is controlled by $\alpha_n$ as $n \to \infty$.

## 5 Empirical Study

In this section, we assess the practical utility of the Gradient-DD method in numerical experiments. To validate performance of Gradient-DD learning, we compare Gradient-DD learning with GTD2 learning, TDC learning (TD with gradient correction (Sutton et al., 2009b)), TDRC learning (TDC with regularized correction (Ghiassian et al., 2020)) and TD learning in both tabular representation and linear representation. We conducted three tasks: a simple random walk task, the Boyan-chain task, and Baird's off-policy counterexample. In each task, we evaluate the performance of a learning algorithm by empirical root mean-squared (RMS) error: $\sqrt{\sum_{s\in\mathcal{S}} d_s(V_{\mathbf{w}_n}(s) - V(s))^2}$. The reason we choose the empirical RMS error rather than root projected mean-squared error or other measures as Ghiassian et al. (2018; 2020) used is because it is a direct measure of concern in practical performance.

### 5.1 Random walk task

As a first test of Gradient-DD learning, we conducted a simple random walk task (Sutton & Barto, 2018). The random walk task has a linear arrangement of $m$ states plus an absorbing terminal state at each end. Thus there are $m + 2$ sequential states, $S_0, S_1, \cdots, S_m, S_{m+1}$, where $m = 10, 20,$ or $40$. Every walk begins in the center state. At each step, the walk moves to a neighboring state, either to the right or to the left with equal probability. If either edge state ($S_0$ or $S_{m+1}$) is entered, the walk terminates. A walk's outcome is defined to be $r = 0$ at $S_0$ and $r = 1$ at $S_{m+1}$. Our aim is to learn the value of each state $V(s)$, where the true values are $(0, 1/(m + 1), \cdots, m/(m + 1), 1)$. In all cases the approximate value function is initialized to the intermediate value $V_0(s) = 0.5$. We consider tabular representation of the value function, as it eliminates the impact of value function approximation. We consider that the learning rate $\alpha_n$ is tapered according to the schedule $\alpha_n = \alpha(10^3 + 1)/(10^3 + n)$. We tune $\alpha \in \{10^{-12/4}, 10^{-11/4}, \cdots, 10^{-1/4}, 1\}$, where larger values were not used since the algorithms tend to diverge when $\alpha$ is large (Fig. 2). We obtain similar results in the case where step sizes are constant (Fig. 6 of the appendix). For GTD2 and TDC, we set $\beta_n = \zeta\alpha_n$ with $\zeta \in \{1/4^5, 1/4^4, 1/4^3, 1/4^2, 1/4, 1, 4, 4^2, 4^3\}$. For Gradient-DD, we set $\kappa = 1$. We also investigate the sensitivity to $\kappa$ in Fig. 7 of the appendix, where we show that $\kappa = 1$ is a good choice in the empirical studies.

To compare algorithms, we begin by plotting the empirical RMS error as a function of step size $\alpha$. To assess convergence performance, we first plot the final empirical RMS error, averaged over the final 100 episodes, as

a function of step size $\alpha$ in the upper panel of Fig. 2. We also plot the average empirical RMS error of all episodes as a function of $\alpha$ and report these results in the upper panel of Fig. 5 of the Appendix. Note that 20,000 episodes are used. From these figures, we can make several observations. (1) Gradient-DD clearly performs better than GTD2. This advantage is consistent in various settings, and gets bigger as the state space becomes large. (3) Gradient-DD performs similarly to TDRC and conventional TD learning, with a similar dependence on $\alpha$, although Gradient-DD exhibits greater sensitivity to the value of $\alpha$ in the log domain than these other algorithms. In addition, Gradient-DD clearly outperforms TDC when $p = 20, 40$ for the range of $\zeta$ that we tested. However, the better comparison is to TD learning since TDC approaches to conventional TD learning when $\zeta$ goes to 0, and TDC appears to offer no advantage over TD learning in this task. In summary, Gradient-DD exhibits clear advantages over the GTD2 algorithm, and its performance is also as good as that of TDRC and conventional TD learning.

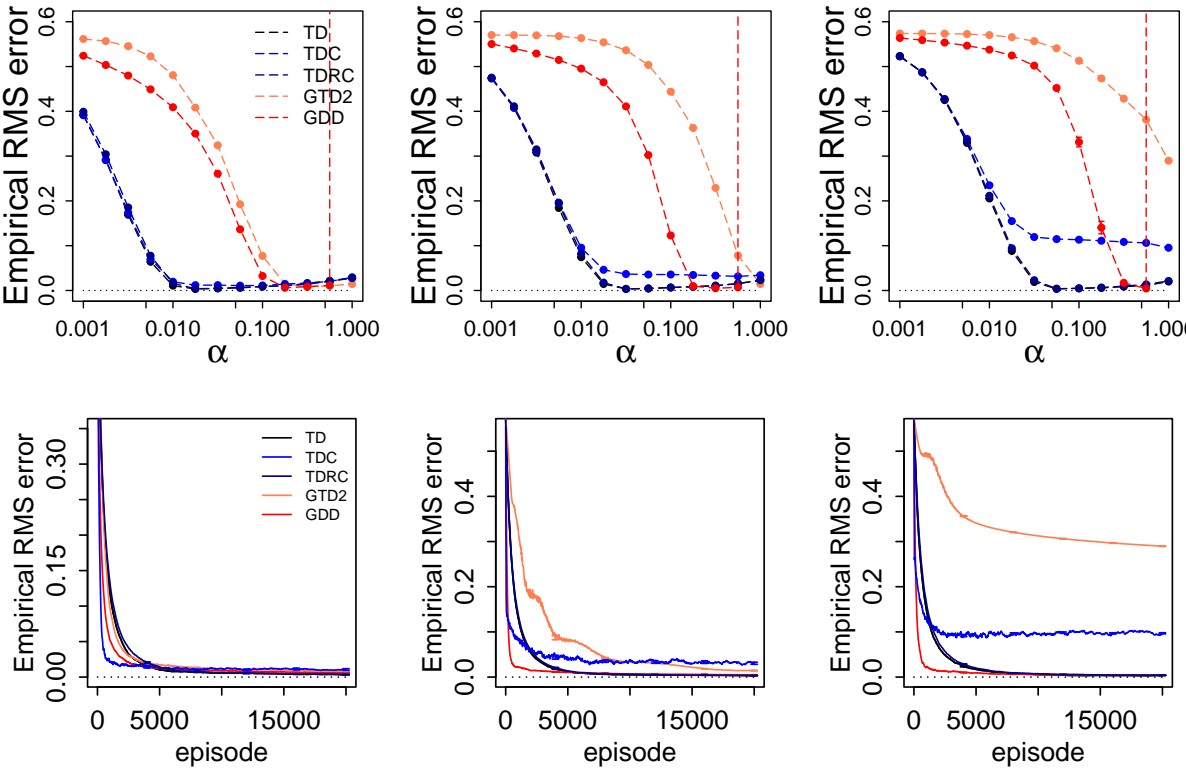

Figure 2: The random walk task with tabular representation and tapering step size $\alpha_n = \alpha(10^3 + 1)/(10^3 + n)$. Upper: Mean error from the final 100 episodes for different values of $\alpha$. Lower: Performance over all episodes, where $\alpha$ is tuned to minimize the mean error from the final 100 episodes. In each row, state space size 10 (left), 20 (middle), or 40 (right). The curves are averaged over 50 runs, with error bars denoting the standard error of the mean, though most are vanishingly small.

Next we look closely at the performance during training in the lower panel of Fig. 2. For each method, we tuned $\alpha \in \{10^{-12/4}, \cdots, 10^{-1/4}, 1\}$ by minimizing the final empirical RMS error, averaged over the last 100 episodes. We also compare the performance when $\alpha$ is tuned by minimizing the average error of all episodes (lower panel of Fig. 5 of the appendix). From these results, we draw several observations. (1) For all conditions tested, Gradient-DD converges much more rapidly than GTD2. The advantage of Gradient-DD grows as the state space increases in size. (2) When evaluating performance based on final episodes and tuning $\alpha$ accordingly (Fig. 2), Gradient-DD exhibits an evidently faster convergence rate than TDRC and conventional TD learning, while demonstrating similar performance in terms of final empirical RMS error. When $\alpha$ is tuned based on the average error of all episodes (Fig. 5), Gradient-DD achieves a slightly smaller error than TDRC and conventional TD learning, even its convergence rate is slighter faster, when the state

space size takes 20 and 40. (3) Gradient-DD has consistent and good performance under both the constant step size setting (Fig. 6) and under the tapered step size setting. In summary, the Gradient-DD learning curves in this task show improvements over other gradient-based methods and performance that matches conventional TD learning.

Like TDRC, the updates of Gradient-DD only have a single shared step size $\alpha_n$, i.e., $\beta_n = \alpha_n$, rather than two independent step sizes $\alpha_n$ and $\beta_n$ as in the GTD2 and TDC algorithms. A possible concern is that the performance gains in our second-order algorithm could just as easily be obtained with existing methods by adopting this two-timescale approach, where the value function weights are updated with a smaller step size than the second set of weights. Hence, in addition to investigating the effects of the learning rate, size of the state space, and magnitude of the regularization parameter, we also investigate the effect of using distinct values for the two learning rates, $\alpha_n$ and $\beta_n$, although we set $\beta_n = \zeta\alpha_n$ with $\zeta = 1$. To do this, we set $\beta_n = \zeta\alpha_n$ for Gradient-DD, with $\zeta \in \{1/64, 1/16, 1/4, 1, 4\}$, and report the results in Fig. 8 of the appendix. The results show that comparably good performance of Gradient-DD is obtained under these various $\zeta$, providing evidence that the second-order difference term in our approach provides an improvement beyond what can be obtained with previous gradient-based methods using the two time scale approach.

## 5.2 Boyan-chain task

We next investigate Gradient-DD learning on the Boyan-chain problem, which is a standard task for testing linear value-function approximation (Boyan, 2002). In this task we allow for $4p - 3$ states, each of which is represented by a $p$-dimensional feature vector, with $p = 20, 50,$ or $100$. The $p$-dimensional representation for every fourth state from the start is $[1, 0, \cdots, 0]$ for state $s_1$, $[0, 1, 0, \cdots, 0]$ for $s_5$, $\cdots$, and $[0, 0, \cdots, 0, 1]$ for the terminal state $s_{4p-3}$. The representations for the remaining states are obtained by linearly interpolating between these. The optimal coefficients of the feature vector are $(-4(p - 1), -4(p - 2), \cdots, 0)/5$. In each state, except for the last one before the end, there are two possible actions: move forward one step or move forward two steps, where each action occurs with probability 0.5. Both actions lead to reward -0.3. The last state before the end just has one action of moving forward to the terminal with reward -0.2. We tune $\alpha \in \{10^{-2}, 10^{-1.5}, 10^{-1}, 10^{-3/4}, 10^{-1/2}, 10^{-1/4}, 10^{-1/8}, 1, 10^{1/8}, 10^{1/4}\}$ for each method by minimizing the average error of the final 100 episodes. All algorithms, with the exception of TD, tend to diverge frequently when $\alpha \geq 10^{1/2}$. TD also experiences frequent divergence when $\alpha \geq 10^{3/4}$. Thus, we set the maximum value of $\alpha$ as $10^{1/4}$. The step size is tapered according to the schedule $\alpha_n = \alpha(2 \times 10^3 + 1)/(2 \times 10^3 + n)$. For GTD2 and TDC, we set $\beta_n = \zeta\alpha_n$ with $\zeta \in \{1/64, 1/16, 1/4, 1, 4\}$. In this task, we set $\gamma = 1$. As in the random-walk task, we set $\kappa = 1$.

We report the performance as a function of $\alpha$ and the performance over episodes in Fig. 3, where we tune $\alpha$ by the performance based on the average error of the last 100 episodes. We also compare the performance based on the average error of all episodes during training and report the results in Fig. 10 of the appendix. These figures lead to conclusions similar to those already drawn in the random walk task. (1) Gradient-DD has much faster convergence than GTD2 and TDC, and generally converges to better values. (However, similar to the random walk task, we note that TDC appears to perform worse than TD learning in this task for nonzero values of $\zeta$, so comparison to TD learning is more informative than comparison to TDC.) (2) Gradient-DD is competitive with TDRC and conventional TD learning despite being somewhat slower at the beginning episodes when $\alpha$ is tuned based on the average error of all episodes. (3) The improvement over GTD2 or TDC grows as the state space becomes larger.

## 5.3 Baird's off-policy counterexample

We also verify the performance of Gradient-DD on Baird's off-policy counterexample (Baird, 1995; Sutton & Barto, 2018), illustrated schematically in Fig. 4, for which TD learning famously diverges. We show results from Baird's counterexample with $N = 7, 20$ states. The reward is always zero, and the agent learns a linear value approximation with $N + 1$ weights $w_1, \cdots, w_{N+1}$: the estimated value of the $j$-th state is $2w_j + w_{N+1}$ for $j \leq N - 1$ and that of the $N$-th state is $w_N + 2w_{N+1}$. In the task, the importance sampling ratio for the dashed action is $(N - 1)/N$, while the ratio for the solid action is $N$. Thus, comparing different state sizes illustrates the impact of importance sampling ratios in these algorithms. The initial weight values are

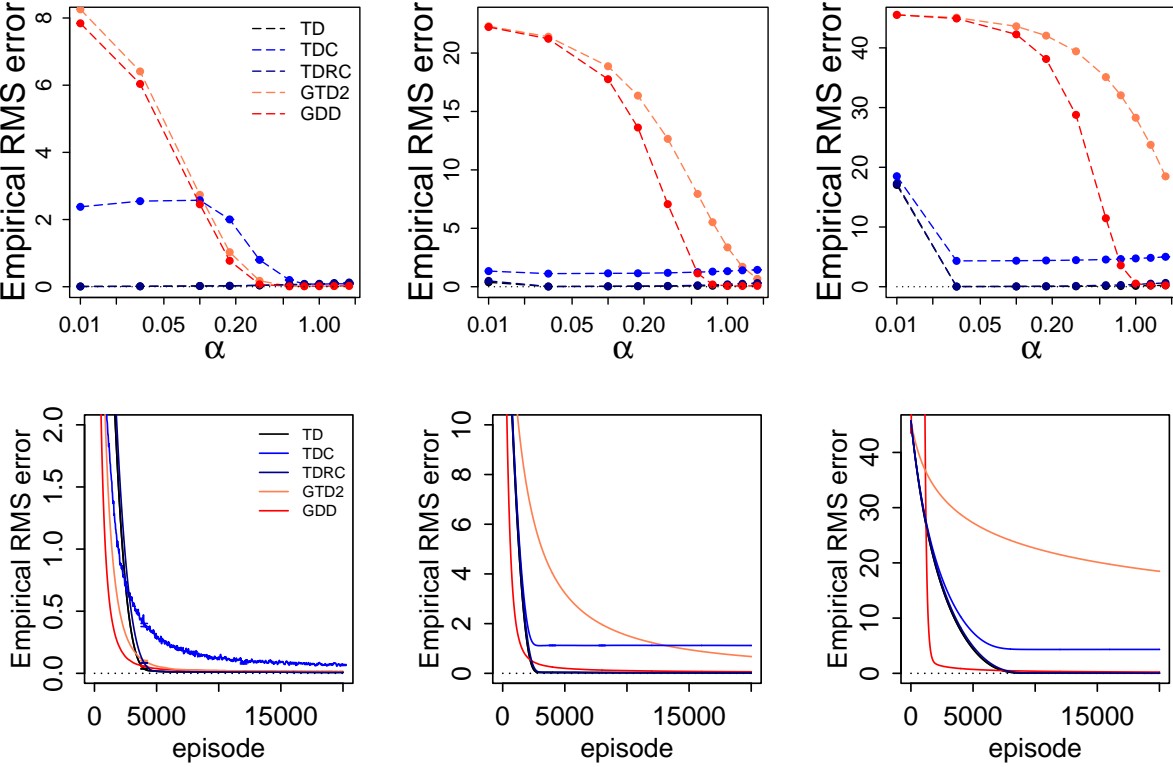

Figure 3: The Boyan Chain task with linear approximation and tapering step size $\alpha_n = \alpha(2 \times 10^3 + 1)/(2 \times 10^3 + n)$. Upper: Performance as a function of $\alpha$; Lower: performance over episodes. In each row, the feature size is 20 (left), 50 (middle), or 100 (right). The curves are averaged over 50 runs, with error bars denoting the standard error of the mean, though most are vanishingly small across runs.

$(1, \cdots, 1, 10, 1)^\top$. Constant $\alpha$ is used in this task and is tuned in the region $\{10^{-16/4}, 10^{-15/4}, \cdots, 10^{-1/4}, 1\}$. We set $\gamma = 0.99$. For TDC and GTD2, thus we tune $\zeta \in \{4^{-2}, 4^{-1}, 1, 4^2, 4^3\}$. Meanwhile we tune $\alpha$ for TDC in a wider region $\{10^{-24/4}, 10^{-23/4}, \cdots, 10^{-1/4}, 1\}$. For Gradient-DD, we tune $\kappa \in \{4^{-1}, 1, 4\}$. We tune $\alpha$ separately for each algorithm by minimizing the average error from the final 100 episodes.

Fig. 4 demonstrates that Gradient-DD works better on this counterexample than GTD2, TDC, and TDRC. It is worthwhile to observe that when the state size is 20, TDRC become unstable, meaning serious unbalance of importance sampling ratios may cause TDRC unstable. We also note that, because the linear approximation leaves a residual error in the value estimation due to the projection error, the RMS errors of GTD2, TDC, and TDRC in this task do not go to zero. In contrast to other algorithms, the errors from our Gradient-DD converge to zero.

## 6 Conclusion and discussion

In this work, we have proposed Gradient-DD learning, a new gradient descent-based TD learning algorithm. The algorithm is based on a modification of the projected Bellman error objective function for value function approximation by introducing a second-order difference term. The algorithm significantly improves upon existing methods for gradient-based TD learning, obtaining better convergence performance than conventional linear TD learning.

Since GTD learning was originally proposed, the Gradient-TD family of algorithms has been extended to incorporate eligibility traces and learning optimal policies (Maei & Sutton, 2010; Geist & Scherrer, 2014), as well as for application to neural networks (Maei, 2011). Additionally, many variants of the vanilla Gradient-TD

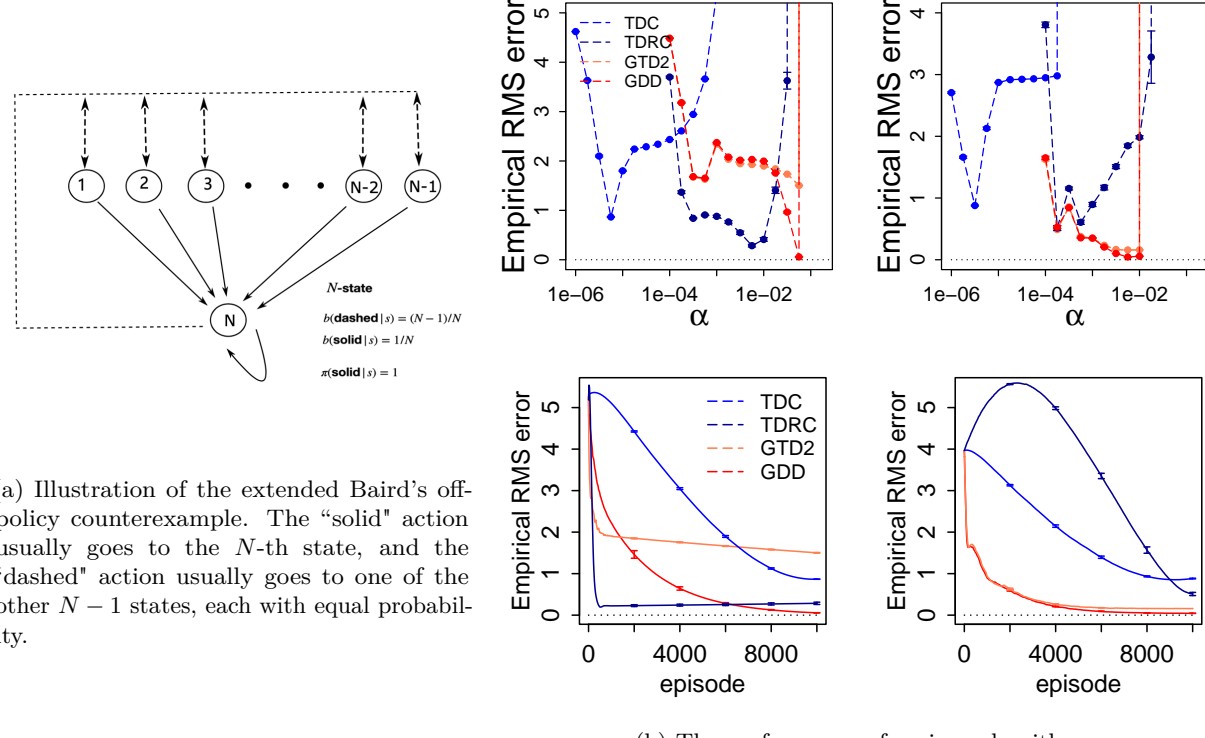

(a) Illustration of the extended Baird's off-policy counterexample. The "solid" action usually goes to the $N$-th state, and the "dashed" action usually goes to one of the other $N-1$ states, each with equal probability.

(b) The performance of various algorithms.

Figure 4: Baird's off-policy counterexample. Upper in (b): Performance as a function of $\alpha$; Lower in (b): performance over episodes. From left to right in (b): 7-state and 20-state.

methods have been proposed, including HTD (Hackman, 2012) and Proximal Gradient-TD (Liu et al., 2016). Because Gradient-DD just modifies the objective error of GTD2 by considering an additional squared-bias term, it may be extended and combined with these other methods, potentially broadening its utility for more complicated tasks.

One potential limitation of our method is that it introduces an additional hyperparameter relative to similar gradient-based algorithms, which increases the computational requirements for hyperparameter optimization. This is somewhat mitigated by our finding that the algorithm's performance is not particularly sensitive to values of $\kappa$, and that $\kappa \sim 1$ was found to be a good choice for the range of environments that we considered. The second limitation lies in the absence of a convergence rate analysis demonstrating superior performance compared to GTD2 in empirical studies, in addition to addressing convergence in this paper. While analyzing asymptotic convergence rates, similar to the approach in Devraj et al. (2019), could be a viable way to assess the proposed Gradient-DD algorithm, such analysis extends beyond the scope of this paper and is deferred to future work. Another potential limitation is that we have focused on value function prediction in the two simple cases of tabular representations and linear approximation. An especially interesting direction for future study will be the application of Gradient-DD learning to tasks requiring more-complex representations, including neural network implementations. Such approaches are especially useful in cases where state spaces are large, and indeed we have found in our results that Gradient-DD seems to confer the greatest advantage over other methods in such cases. Intuitively, we expect that this is because the difference between the optimal update direction and that chosen by gradient descent becomes greater in higher-dimensional spaces (cf. Fig. 1). This performance benefit in large state spaces suggests that Gradient-DD may be of practical use for these more challenging cases.

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

# A  Appendix

## A.1  On the equivalence of Eqns. (8) & (9)

The Karush-Kuhn-Tucker conditions of Eqn. (9) are the following system of equations

$$
\begin{cases}
\frac{d}{d\mathbf{w}} J(\mathbf{w}) + \kappa \frac{d}{d\mathbf{w}} (\|\mathbf{V_w} - \mathbf{V_{w_{n-1}}}\|_{\mathbf{D}}^2 - \mu) = 0; \\
\kappa (\|\mathbf{V_w} - \mathbf{V_{w_{n-1}}}\|_{\mathbf{D}}^2 - \mu) = 0; \\
\|\mathbf{V_w} - \mathbf{V_{w_{n-1}}}\|_{\mathbf{D}}^2 \le \mu; \\
\kappa \ge 0.
\end{cases}
$$

These equations are equivalent to

$$
\begin{cases}
\frac{d}{d\mathbf{w}} J(\mathbf{w}) + \kappa \frac{d}{d\mathbf{w}} \|\mathbf{V_w} - \mathbf{V_{w_{n-1}}}\|_{\mathbf{D}}^2 = 0 \text{ and } \kappa > 0, \\
\qquad\qquad\qquad\qquad\qquad \text{if } \|\mathbf{V_w} - \mathbf{V_{w_{n-1}}}\|_{\mathbf{D}}^2 = \mu; \\
\frac{d}{d\mathbf{w}} J(\mathbf{w}) = 0 \text{ and } \kappa = 0, \text{ if } \|\mathbf{V_w} - \mathbf{V_{w_{n-1}}}\|_{\mathbf{D}}^2 < \mu.
\end{cases}
$$

Thus, for any $\mu > 0$, there exists a $\kappa \ge 0$ such that $\frac{d}{d\mathbf{w}} J(\mathbf{w}) + \mu \frac{d}{d\mathbf{w}} \|\mathbf{V_w} - \mathbf{V_{w_{n-1}}}\|_{\mathbf{D}}^2 = 0$.

## A.2  The relation to natural gradients

In this section, we shall show that Gradient-DD is related to, but distinct from, the natural gradient. We thank a reviewer for pointing out the connection between Gradient-DD and the natural gradient.

Following Amari (1998) or Thomas (2014), the natural gradient of $J(\mathbf{w})$ is the direction obtained by solving the following optimization:

$$
\lim_{\epsilon \to 0} \arg \min_{\Delta} J(\mathbf{w} + \epsilon \Delta) \text{ s.t. } \epsilon^2 \Delta^\top \mathbf{X}^\top \mathbf{D} \mathbf{X} \Delta \le \mu. \tag{A.1}
$$

We can note that this corresponds to the ordinary gradient in the case where the metric tensor $\mathbf{X}^\top \mathbf{D} \mathbf{X}$ is proportional to the identity matrix.

Now we rewrite Eqn. (9) as

$$
\|\mathbf{V_w} - \mathbf{V_{w_{n-1}}}\|_{\mathbf{D}}^2 = (\mathbf{w} - \mathbf{w}_{n-1})^\top \mathbf{X}^\top \mathbf{D} \mathbf{X} (\mathbf{w} - \mathbf{w}_{n-1}).
$$

Denote $\epsilon \Delta = \mathbf{w} - \mathbf{w}_{n-1}$, where $\epsilon$ is the radius of the circle of $\mathbf{w}$ around $\mathbf{w}_{n-1}$ and $\Delta$ is a unit vector. Thus, we have

$$
\|\mathbf{V_w} - \mathbf{V_{w_{n-1}}}\|_{\mathbf{D}}^2 = \epsilon^2 \Delta^\top \mathbf{X}^\top \mathbf{D} \mathbf{X} \Delta.
$$

For the MSPBE objective, we have

$$
J(\mathbf{w}) = J(\mathbf{w}_{n-1} + \mathbf{w} - \mathbf{w}_{n-1}) = J(\mathbf{w}_{n-1} + \epsilon \Delta).
$$

Minimizing Eqn. (9) is equivalent to the following optimization

$$
\arg \min_{\Delta} J(\mathbf{w}_{n-1} + \epsilon \Delta) \text{ s.t. } \epsilon^2 \Delta^\top \mathbf{X}^\top \mathbf{D} \mathbf{X} \Delta \le \mu. \tag{A.2}
$$

In the limit as $\epsilon \to 0$, the above optimization is equivalent to

$$
\arg \min_{\Delta} \Delta^\top \nabla J(\mathbf{w}_{n-1}) \text{ s.t. } \epsilon^2 \Delta^\top \mathbf{X}^\top \mathbf{D} \mathbf{X} \Delta \le \mu.
$$

Thus, given the metric tensor $\mathbf{G} = \mathbf{X}^\top \mathbf{D} \mathbf{X}$, $-\mathbf{G}^{-1} \nabla J(\mathbf{w}_{n-1})$ is the direction of steepest descent, *i.e.* the natural gradient, of $J(\mathbf{w}_{n-1})$. The natural gradient of $J(\mathbf{w})$, on the other hand, is the direction of steepest descent at $\mathbf{w}$, rather than at $\mathbf{w}_{n-1}$.

Therefore, our Gradient-DD approach makes use of the natural gradient of the objective around $\mathbf{w}_{n-1}$ rather than around $\mathbf{w}_n$ in Eqn. (A.1). This explains the distinction of the updates of our Gradient-DD approach from the updates of directly applying the natural gradient of the objective $\mathbf{w}$.

### A.3 Proof of Theorem 1

We introduce an ODE result on stochastic approximation in the following lemma, then show the convergence of our Gradient-DD approach in Theorem 1 by applying this result.

**Lemma 1.** *(Theorems 2.1 & 2.2 of (Borkar & Meyn, 2000)) Consider the stochastic approximation algorithm described by the d-dimensional recursion*

$$\mathbf{y}_{n+1} = \mathbf{y}_n + a_n[f(\mathbf{y}_n) + \mathbf{M}_{n+1}].$$

*Suppose the following conditions hold: (c1) The sequence $\{\alpha_n\}$ satisfies $0 < \alpha_n < 1$, $\sum_{n=1}^{n} \alpha_n = \infty$, $\sum_{n=1}^{n} \alpha_n^2 < \infty$. (c2) The function $f$ is Lipschitz, and there exists a function $f_\infty$ such that $\lim_{r \to \infty} f_r(\mathbf{y}) = f_\infty(\mathbf{y})$, where the scaled function $f_r : \mathbb{R}^d \to \mathbb{R}^d$ is given by $f_r(\mathbf{y}) = f(r\mathbf{y})/r$. (c3) The sequence $\{\mathbf{M}_n, \mathcal{F}_n\}$, with $\mathcal{F}_n = \sigma(\mathbf{y}_i, \mathbf{M}_i, i \leq n)$, is a martingale difference sequence. (c4) For some $c_0 < \infty$ and any initial condition $y_0$, $E(\|\mathbf{M}_{n+1}\|^2|\mathcal{F}_n) \leq c_0(1 + \|\mathbf{y}_n\|^2)$. (c5) The ODE $\dot{\mathbf{y}} = f_\infty(\mathbf{y})$ has the origin as a globally asymptotically stable equilibrium. (c6) The ODE $\dot{\mathbf{y}}(t) = f(\mathbf{y}(t))$ has a unique globally asymptotically stable equilibrium $\mathbf{y}^*$. Then (1) under the assumptions (c1-c5), $\sup_n \mathbf{y}_n < \infty$ in probability. (2) under the assumptions (c1-c6), as $n \to \infty$, $\mathbf{y}_n$ converges to $\mathbf{y}^*$ with probability 1 .*

Now we investigate the stochastic gradient descent updates in Eqn. (13), which is recalled as follows:

$$\boldsymbol{\rho}_{n+1} = \boldsymbol{\rho}_n - \kappa\alpha_n\mathbf{H}_n(\boldsymbol{\rho}_n - \boldsymbol{\rho}_{n-1}) + \alpha_n(\mathbf{G}_n\boldsymbol{\rho}_n + \mathbf{g}_{n+1}). \tag{A.3}$$

The iterative process in Eqn. (A.3) can be rewritten as

$$(\boldsymbol{\rho}_{n+1} - \boldsymbol{\rho}_n) = -\kappa\alpha_n\mathbf{H}_n(\boldsymbol{\rho}_n - \boldsymbol{\rho}_{n-1}) + \alpha_n(\mathbf{G}_n\boldsymbol{\rho}_n + \mathbf{g}_{n+1}). \tag{A.4}$$

Defining

$$\mathbf{u}_{n+1} = \boldsymbol{\rho}_{n+1} - \boldsymbol{\rho}_n.$$

Eqn. (A.4) becomes

$$\mathbf{u}_{n+1} = -\kappa\alpha_n\mathbf{H}_n\mathbf{u}_n + \alpha_n(\mathbf{G}_n\boldsymbol{\rho}_n + \mathbf{g}_{n+1}).$$

Thus, the iterative process in Eqn. (A.3) is rewritten as two parallel processes that are given by

$$\boldsymbol{\rho}_{n+1} = \boldsymbol{\rho}_n - \kappa\alpha_n\mathbf{H}_n\mathbf{u}_n + \alpha_n(\mathbf{G}_n\boldsymbol{\rho}_n + \mathbf{g}_{n+1}), \tag{A.5}$$
$$\mathbf{u}_{n+1} = -\kappa\alpha_n\mathbf{H}_n\mathbf{u}_n + \alpha_n(\mathbf{G}_n\boldsymbol{\rho}_n + \mathbf{g}_{n+1}). \tag{A.6}$$

Our proofs have three steps. First we shall show $\sup_n \|\boldsymbol{\rho}_n\|$ is bounded by applying the ordinary differential equation approach of the stochastic approximation (the 1st result of Lemma 1) into the recursion Eqn. (A.5). Second, based on this result, we shall show that $\mathbf{u}_n$ goes to 0 in probability by analyzing the recursion Eqn. (A.6). At last, along with the result that $\mathbf{u}_n$ goes to 0 in probability, by applying the 2nd result of Lemma 1 into the recursion Eqn. (A.5), we show that $\boldsymbol{\rho}_n$ goes to the TD fixed point, which is given by the solution of $\mathbf{G}\boldsymbol{\rho} + \mathbf{g} = 0$.

First, we shall show $\sup_n \|\boldsymbol{\rho}_n\|$ is bounded. Eqn. (A.5) is rewritten as

$$\boldsymbol{\rho}_{n+1} = \boldsymbol{\rho}_n + \alpha_n(f(\boldsymbol{\rho}_n) + \mathbf{M}_{n+1}), \tag{A.7}$$

where $f(\boldsymbol{\rho}_n) = (\mathbf{G}\boldsymbol{\rho}_n + \mathbf{g}) - \kappa\mathbf{H}\mathbf{u}_n$ and $\mathbf{M}_{n+1} = ((\mathbf{G}_n - \mathbf{G})\boldsymbol{\rho}_n + \mathbf{g}_{n+1} - \mathbf{g}) - \kappa(\mathbf{H}_n - \mathbf{H})\mathbf{u}_n$. Let $\mathcal{F}_n = \sigma(\mathbf{u}_0, \boldsymbol{\rho}_0, \mathbf{M}_0, \mathbf{u}_1, \boldsymbol{\rho}_1, \mathbf{M}_1, \cdots, \mathbf{u}_n, \boldsymbol{\rho}_n, \mathbf{M}_n)$ be $\sigma$-fields generated by the quantities $\mathbf{u}_i, \boldsymbol{\rho}_i, \mathbf{M}_i, i \leq n$.

Now we verify the conditions (c1-c5) of Lemma 1. Condition (c1) is satisfied under the assumption of the step sizes. Clearly, $f(\mathbf{u})$ is Lipschitz and $f_\infty(\boldsymbol{\rho}) = \mathbf{G}\boldsymbol{\rho}$, meaning Condition (c2) is satisfied. Condition (c3) is satisfied by noting that $(M_n, \mathcal{F}_n)$ is a martingale difference sequence, i.e., $E(\mathbf{M}_{n+1}|\mathcal{F}_n) = 0$.

We next investigate $E(\|\mathbf{M}_{n+1}\|^2|\mathcal{F}_n)$. From the triangle inequality, we have that

$$\|\mathbf{M}_{n+1}\|^2 \leq 2\|(\mathbf{G}_n - \mathbf{G})\|^2\|\boldsymbol{\rho}_n\|^2 + 2\|\kappa(\mathbf{H}_n - \mathbf{H})\|^2\|\mathbf{u}_n\|^2. \tag{A.8}$$

From Assumption A3 in Theorem 1 that $\|\mathbf{u}_n\|$ is bounded and the Assumption A1 that $(\mathbf{x}_n, r_n, \mathbf{x}_{n+1})$ is an i.i.d. sequence with uniformly bounded second moments, there exists some constant $c_0$ such that

$$\|\mathbf{G}_n - \mathbf{G}\|^2 \leq c_0/2, \text{ and } \|\kappa(\mathbf{H}_n - \mathbf{H})\|^2\|\mathbf{u}_n\|^2 \leq c_0/2.$$

Thus, Condition (c4) is satisfied.

Note that $\mathbf{G}$ is defined in (Maei, 2011). From (Sutton et al., 2009a) and (Maei, 2011), the eigenvalues of the matrix $\mathbf{G}$ are strictly negative under the Assumption A2. Therefore, Condition (c5) is satisfied. Thus, applying the 1st part of Lemma 1 shows that $\sup_n \|\boldsymbol{\rho}_n\|$ is bounded in probability.

Second, we investigate the recursion Eqn. (A.6). Let $\mathbf{y}_{n+1} = (\mathbf{G}_n\boldsymbol{\rho}_n + \mathbf{g}_{n+1})$. Then

$$\begin{aligned}
\mathbf{u}_{n+1} &= \alpha_n[-\kappa\mathbf{H}_n\mathbf{u}_n + \mathbf{y}_{n+1}] \\
&= \alpha_n\mathbf{y}_{n+1} + \alpha_n\alpha_{n-1}(-\kappa\mathbf{H}_n)\mathbf{y}_n + \alpha_n\alpha_{n-1}\alpha_{n-2}(-\kappa\mathbf{H}_n)(-\kappa\mathbf{H}_{n-1})\mathbf{y}_{n-1} \\
&\quad + \cdots + \alpha_n\prod_{k=0}^{n-1}\alpha_k(-\kappa\mathbf{H}_{k+1})\mathbf{y}_1 + \prod_{k=0}^{n}\alpha_k(-\kappa\mathbf{H}_k)\mathbf{u}_0.
\end{aligned} \tag{A.9}$$

Note that $\|\mathbf{H}_n\| \leq 1/\kappa$ due to $\|\mathbf{x}_n\| \leq 1/\kappa$ and that there exists a constant $c$ such that $\|\boldsymbol{\rho}_n\| \leq c$ due to the above result that $\sup_n \|\boldsymbol{\rho}_n\| < \infty$ in probability. Without loss of generality, we assume that $\|\mathbf{x}_n\| \leq 1/\kappa$. Eqn. (A.9) implies that

$$\|\mathbf{u}_{n+1}\| \leq c\left(\alpha_n + \alpha_n\alpha_{n-1} + \alpha_n\alpha_{n-1}\alpha_{n-2} + \cdots + \prod_{k=0}^{n}\alpha_k\right) + \prod_{k=0}^{n}\alpha_k\|\mathbf{u}_0\|. \tag{A.10}$$

Under Assumption A0, $\alpha_n \to 0$ as $n \to 0$. Based on this, Lemma 2 (given in the following section) tells us that $\alpha_n + \alpha_n\alpha_{n-1} + \alpha_n\alpha_{n-1}\alpha_{n-2} + \cdots + \prod_{k=0}^{n}\alpha_k \to 0$ as $n \to 0$. Thus, Eqn. (A.10) implies that $\mathbf{u}_n \to 0$ in probability.

Finally, for applying the 2nd part of Lemma 1, we just need to verify Condition (c6). Because $\mathbf{u}_n$ goes to 0 with probability 1, Eqn. (A.7) tells us that the associated ODE corresponds to

$$\mathbf{G}\boldsymbol{\rho} + \mathbf{g} = 0.$$

Thus, Condition (c6) is satisfied. The theorem is proved.

### A.4  A Lemma

**Lemma 2.** *Denote $\epsilon_n = \alpha_n + \alpha_n\alpha_{n-1} + \cdots + \alpha_n\alpha_{n-1}\cdots\alpha_0$. If $\alpha_n \to 0$ as $n \to \infty$, then $\epsilon_n \to 0$ as $n \to \infty$.*

*Proof.* Because $\alpha_n \to 0$ as $n \to \infty$, there exists $\alpha \in (0,1)$ and some integer $N$ such that $\alpha_n \leq \alpha < 1$ when $n \geq N$. Define a sequence $\varepsilon_n$ such that

$$\varepsilon_n = 1 + \alpha\varepsilon_{n-1} \text{ for } n \geq N+1;$$
$$\varepsilon_N = \epsilon_N.$$

Obviously,

$$\epsilon_n \leq \varepsilon_n, \forall n \geq N. \tag{A.11}$$

Now we investigate the sequence $\varepsilon_n$.

$$\begin{aligned}
\varepsilon_n &= 1 + \alpha\varepsilon_{n-1} = 1 + \alpha(1 + \alpha\varepsilon_{n-2}) = \cdots = 1 + \alpha + \cdots + \alpha^{n-N-1} + \alpha^{n-N}\varepsilon_N \\
&\leq \sum_{k=0}^{\infty}\alpha^k + \alpha^{n-N}\varepsilon_N = 1/(1-\alpha) + \alpha^{n-N}\varepsilon_N.
\end{aligned}$$

Thus, we have that

$$\sup_{n \geq N}\varepsilon_n < \infty. \tag{A.12}$$

From Eqns. (A.11) & (A.12), we have

$$\sup_{n \geq 0} \epsilon_n < \infty. \tag{A.13}$$

From the definition of $\epsilon_n$, we have that $\epsilon_n = \alpha_n + \alpha_n \epsilon_{n-1}$. It follows that

$$\alpha_n = \frac{\epsilon_n}{1 + \epsilon_{n-1}} \geq \frac{\epsilon_n}{1 + \sup_{k \geq 0} \epsilon_k}.$$

From the assumption $\alpha_n \to 0$ as $n \to \infty$ and Eqn. (A.13), we have $\epsilon_n \to 0$ as $n \to \infty$. □

## A.5  Additional empirical results

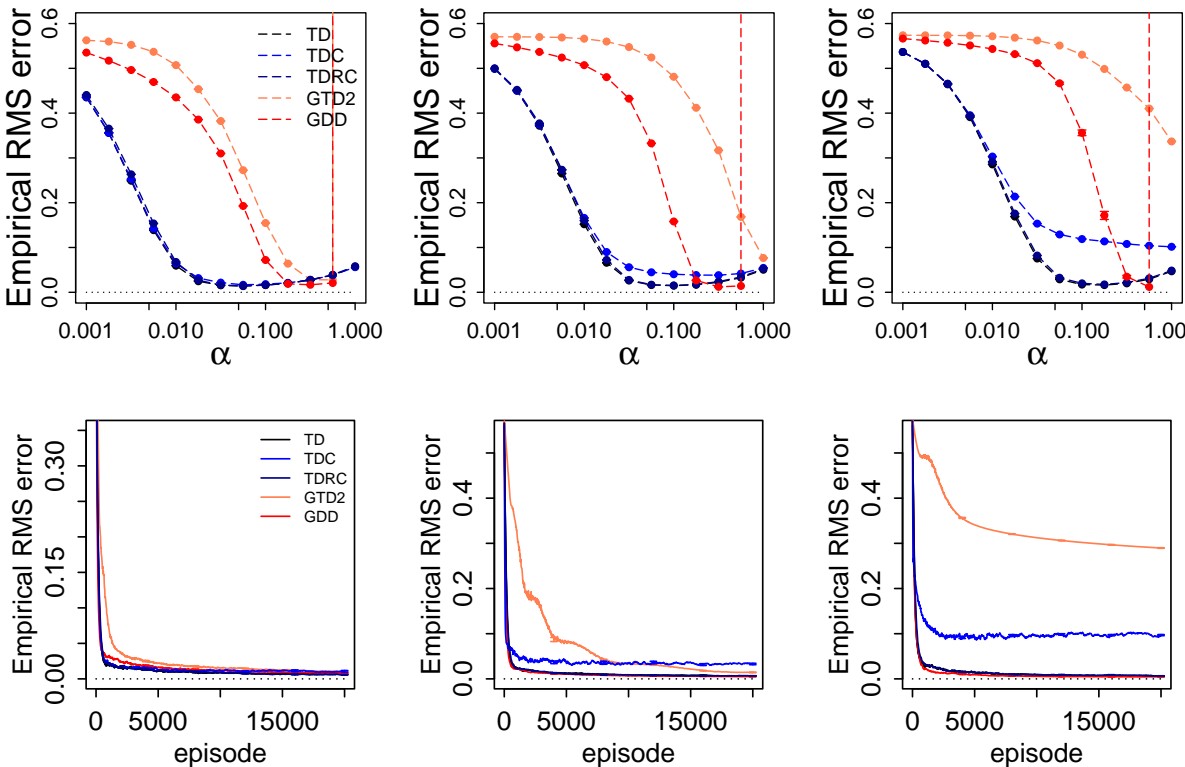

Figure 5: The random walk task with tabular representation. The setting is similar to Fig. 2, but the performance is evaluated by the average error of all episodes, and $\alpha$ is tuned by minimizing the average error of all episodes. Upper: Performance as a function of $\alpha$; Lower: performance over episodes. From left to right: state space size 10 (left), 20 (middle), or 40 (right).

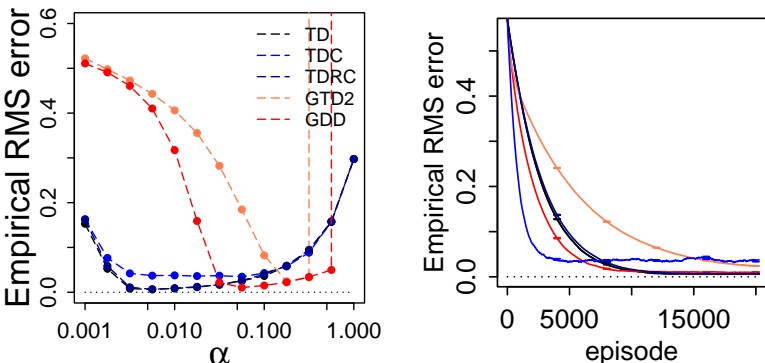

Figure 6: The random walk task with the tabular representation. The setting is similar to Fig. 2, but constant step size is used. The state size is 20. The curves are averaged over 50 runs, with error bars denoting the standard error of the mean, though most are vanishingly small.

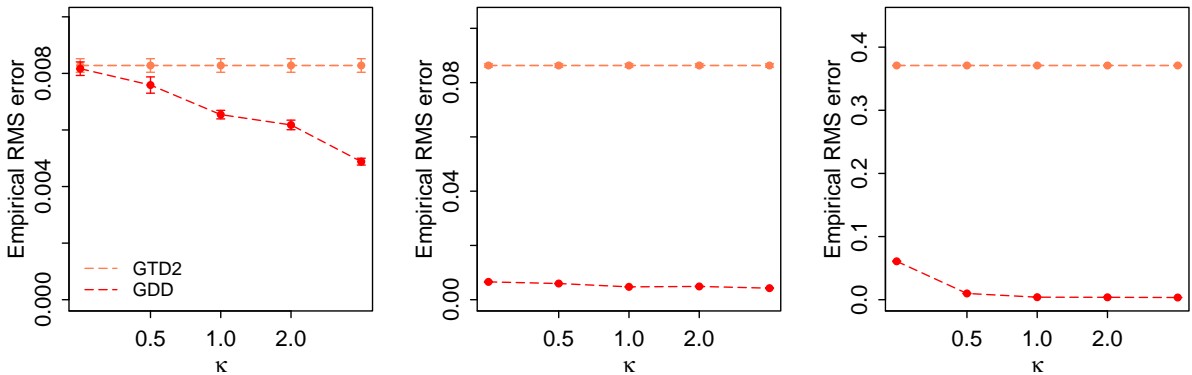

Figure 7: Performance of Gradient-DD in the random walk task in the tabular representation with $\kappa \in \{0.25, 0.5, 1, 2, 4\}$. From left to right: state space size 10 (left), 20 (middle), or 40 (right). In each figure, $\alpha$ is tuned for each algorithm by minimizing the average error of the last 100 episodes. Results are averaged over 50 runs, with error bars denoting standard error of the mean.

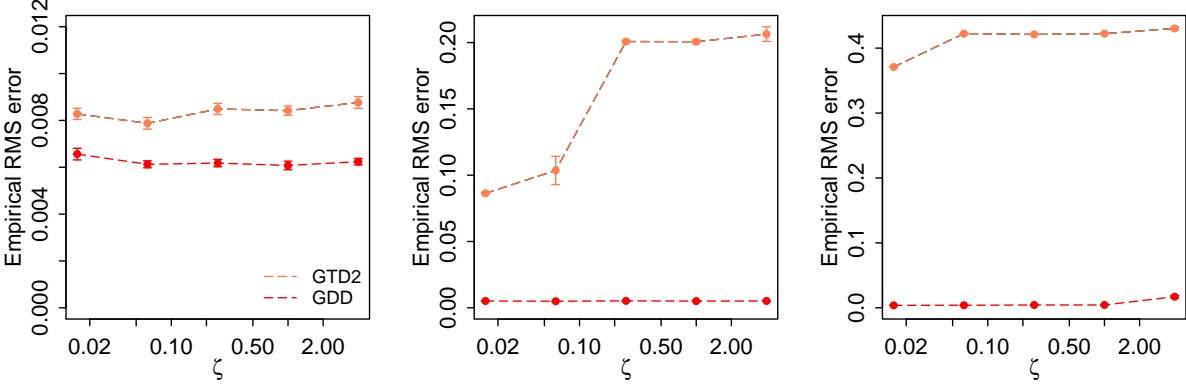

Figure 8: The random walk task in the tabular representation. Performance for various $\beta_n = \zeta\alpha_n$, with $\zeta \in \{4^{-3}, 4^{-2}, 4^{-1}, 1, 4\}$. From left to right in each row: the size of the state space is $m = 10$, $m = 20$, and $m = 40$. In each case $\alpha$ is tuned by minimizing the average error of the last 100 episodes according to the their performance of corresponding algorithms. Results are averaged over 50 runs, with error bars denoting standard error of the mean.

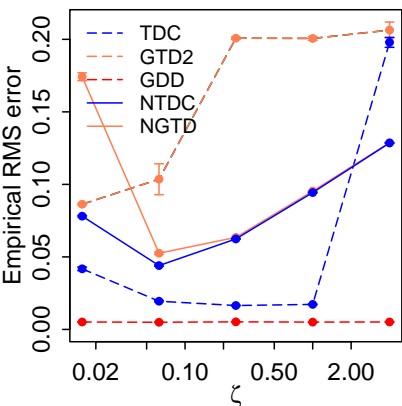

Figure 9: Performance of natural TDC and natural GTD2 in the random walk task with the tabular representation and $m = 20$. Performance for various $\beta_n = \zeta\alpha_n$, with $\zeta \in \{4^{-3}, 4^{-2}, 4^{-1}, 1, 4\}$. In each case $\alpha$ is tuned by minimizing the average error of the last 100 episodes according to the their performance of corresponding algorithms. "NGTD" and "NTDC" denote the natural gradient-based algorithm of GTD2 and TDC, respectively. Results are averaged over 50 runs, with error bars denoting standard error of the mean.

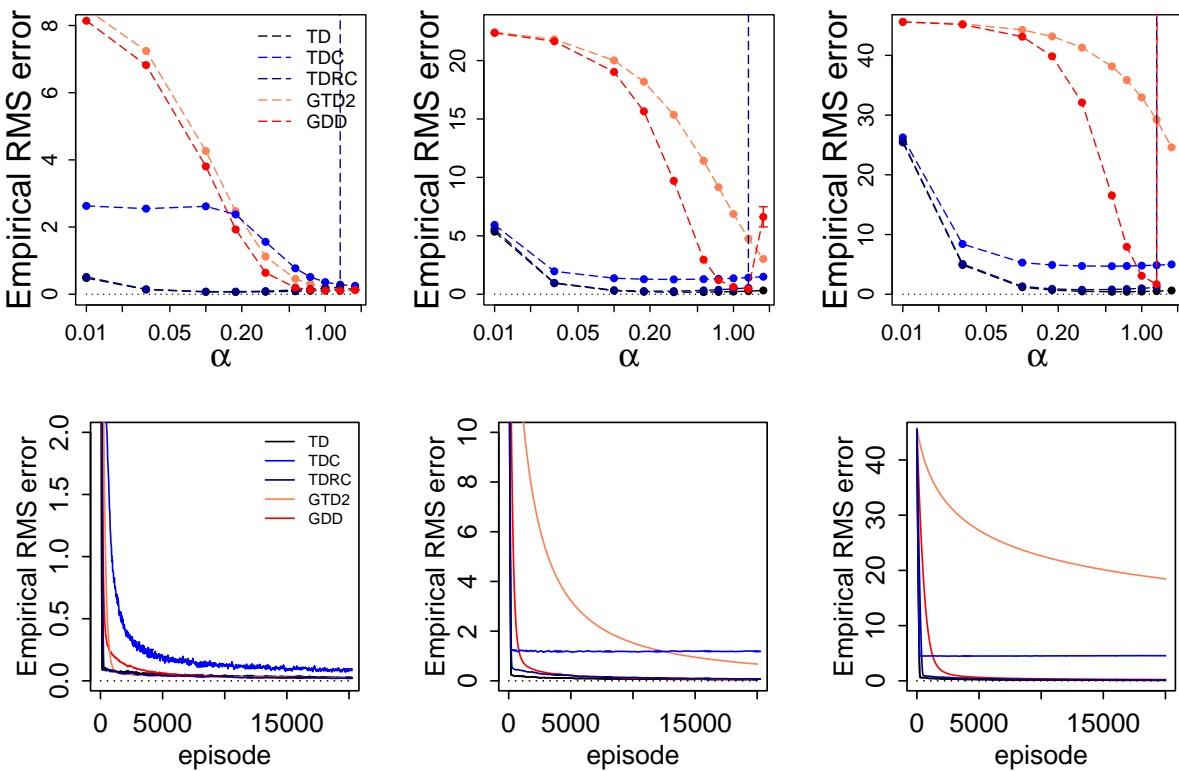

Figure 10: The Boyan Chain task with linear approximation. The setting is similar to Fig. 3, but we evaluate the performance the average error of all episodes, and $\alpha$ is tuned by minimizing the average error of all episodes. Upper: Performance as a function of $\alpha$; Lower: performance over episodes.

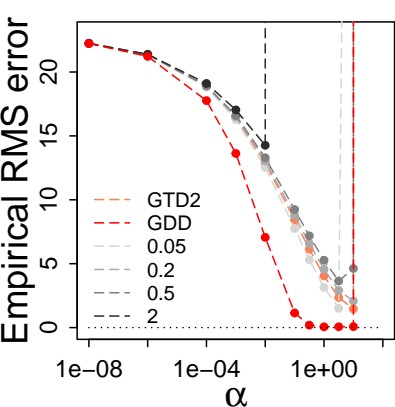

Figure 11: Results from the Boyan-chain task with a feature size of 50 when regularizing the objective in the parameter space, unlike Gradient-DD, which regularizes in the space of the value function. The various shades of gray represent different values of the regularization parameter $\kappa$ in this regularization scheme.

