# OpenReview forum: "Gradient Descent Temporal Difference-difference Learning"
_TMLR — Rejected by TMLR_

### Review · Reviewer_9XJi · 2023-10-21

**Summary Of Contributions:**

This work focuses on policy evaluation with linear function approximation in reinforcement learning. The work builds upon the gradient-temporal-difference (GTD) class of methods by modifying the objective function, specifically adding an error term for the distance between consecutive estimates of the value-function. Then, the work proposes a stochastic approximation algorithm to solve the proposed objective function. The algorithm is theoretically proven to converge to a unique solution and experimentally shown to outperform other existing methods, particularly of the GTD class.

**Audience:**

Yes

**Broader Impact Concerns:**

I don't foresee concerns regarding the broader impact of this work.

**Claims And Evidence:**

Yes

**Requested Changes:**

My suggestions for changes are not critical to secure recommendation for acceptance. The suggestionsthey mostly regard the exposition of the text, and not the technical correctness. Moreover, my suggestions should be easily integrated by the authors on the paper.

- I suggest that the motivation for the work, i.e. the first paragraph of the introduction, drops the focus on the fact that learning can happen "off-policy". In fact, the off-policy notion of learning does not appear in the rest of the document, so I felt this emphasis a bit strange. I suggest to start by motivating with RL itself, and the policy evaluation problem in general, and then possibly emphasize the off-policy aspect (but if that is emphasized, it must be further discussed throughout the text).
- Still related to the "off-policy" comment above, the state $s_{n+1}$ is used on the target for the update of $w_n$ and also as the basis for the update of $w_{n+1}$, i.e., the data is following online, why is the algorithm performing off-policy learning? Is it simply because the projection operator considers a matrix $D$ that weighs the states by their probability? If the data is coming online, those probabilities will be the ones induced by the policy on the Markov decision process, i.e. of the corresponding Markov chain. This should be clarified.
- On the related work, it is important to, at least briefly, explain what is the natural gradient and the natural-gradient algorithms if we want the reader to be able to understand the comparison. Currently, the text only says what the natural gradient is not, rather then what it actually is, and it is difficult to understand the comparison with the ideas proposed on the paper;
- In section 2.2, I think that R and P that appear on the Bellman equation should depend on the target policy, not on the behavior policy as is written;
- In section 2.2, matrix D has the probabilities of state-visitation under the target or behavior policy? I think it is the behavior one, but it should be clarified;
- In section 2.2, some additional references would be important. for instance cite "Reinforcement Learning: An Introduction", Sutton and Barto, for the notion that minimizing the Bellman error does not provide in general a good approximation of the value-function; cite the original GTD methods precisely when introducing the objective function $J$;
- In section 2.2, the objective function $J$ is instatiated at $w$ (equation (5)) but the expression depends on $w_n$. Further, in Equation (9) of section 3, a similar problem happen. Furthermore, I think the expression for the gradient only makes sense if the data is i.i.d., otherwise the objective function and the gradient must depend not only on the parameters but also on $n$;
- In section 3, it is a bit strange to write $k \sim O(1)$ instead of simply saying that $k$ is constant;
- In section 4, the convergence analysis could be motivated more clearly by saying that the section will prove that the added term to the objective and the method do not harm the asymptotic convergence guarantees of the original GTD methods, nor the limit solution. As is, it the reader may tend to think that the previous methods did not have convergence guarantees and this one does;
- The experiments are in general very satisfactory. I would explain better the baselines (TDC and the others), at least in the appendix;
- In the first paragraph of section 3.1, on the Random Walk Task, if there are $m+2$ states, how is the value $(1, ..., M)/(m+1)$? What are the the values of the extreme cases, even though they are simple to calculate? They should appear;
- Throughout the analysis of the experiments, the benefits of the proposed method are understated. As far as I can see, the proposed method is outperforming TD-learning and TDRC, not only "as good as". Why do the authors understand their method is not actually outperforming conventional TD-learning and conventional TDRC? This could be either altered or better explained;
- It could be also interesting to see in the experiments if simply regularizing the objective by $w_n - w_{n-1}$, instead of $V_n - V_{n-1}$ would also provide similar results. Alternatively, to see if techniques of gradient clipping would provide similar results. This suggestion is not major.
- In the conclusion, there is a reference to the benefits of applying the techniques introduced in this work to the case of non-linear function approximation. Are these results in appendix, or somewhere available? If not, I don't believe they should be mentioned, as they will not be accessible nor reproducible;
-  In the appendix, there is a typo "by by", instead of simply "by".

**Strengths And Weaknesses:**

The paper for this work is well-written, clear and easy to follow. Specifically, the problem is mostly well motivated, the related work is well-contrasted and the contributions are clearly explained. Then, the experimental validation of the proposed method, which is novel, does not leave doubt regarding its benefits over the GTD class of methods. Moreover, the contributions should be interesting to the reinfrocement learning (RL) community around the more practical-side of the convergence problem of RL with function approximation. Finally, the paper has potential to influence future work by bringing ideas that were known to be benefitial in other settings (a trust region in the improvement step in the case of the sota PPO algorithm , for example) to a simpler setting and analyzing them thereat.

On the downside, I think the exposition of the paper, specifically in the introduction, experimental discussion and conclusion, could have been more inflated. It is my understanding that, since contributions of the work are quite interesting and well-explained, the authors did not need to understate some points.

---

> ### Author Response · Authors · 2023-12-29
>
> Thanks to this reviewer for numerous constructive comments on how to improve our manuscript. We have addressed each comment, with the the significant changes described below.
>
> **The motivation of the paper**
>
> Following the reviewer's suggestion, the first two paragraphs in the introduction and the abstract have been extensively rewritten. In this revised version, the motivation is centered around gradient-based methods as alternatives to semi-gradient methods.
>
> **The notation of the general setting**
>
> Section 2.2 has been modified in response to the reviewer's comments. In this revision, the general setting has been adapted accordingly, and we have introduced the notation for off-policy learning in GTD2 in Section 2.2 and Gradient-DD in Section 3.
>
> **The motivation of the convergence analysis**
>
> Following the reviewer's suggestion, the motivation for the convergence analysis has been clarified, as reflected in the first paragraph of Section 4.
>
> **On the experimental discussion and conclusion**
>
> We have included two detailed conclusions as follows:
> (1) We have discussed both settings: evaluating performance based on final episodes (the figures in the main text) and evaluating performance based on all episodes (the figures in the appendix).
> (2) We have used wider region $\zeta$ for optimizing the performance of TDC since it will get close to TD when $\zeta$ is very small. The conclusion has been updated accordingly.
>
> **The penalty in the parameter space**
>
> We conducted tests to understand the implications of placing the penalty in the parameter space, as opposed to Gradient-DD's approach of placing the penalty in the space of the value function.
> Refer to the highlighted paragraph in Section 3 and Figure 11 in the appendix.

---

### Review · Reviewer_N9ri · 2023-10-23

**Summary Of Contributions:**

The paper proposes a new GTD2-like GDD algorithm for value function approximation, via solving the original MSPBE problem. The main difference between GDD and GTD2 algorithm seems to be a modification of the update rule for the "dual variable". Specifically, this "modified" update rule is something I would consider a momentum version of the original update rule that appears in GTD2. The authors prove convergence under finite state and action space setting of this new GDD algorithm, and also show via empirical results that the algorithm is faster to converge.

**Audience:**

Yes

**Broader Impact Concerns:**

I am not concerned of any broader impact concerns from this work.

**Claims And Evidence:**

Yes

**Requested Changes:**

At-least better numerical results are required. Additional changes are suggested based the weaknesses above.

**Strengths And Weaknesses:**

Strengths:

1. The paper is well-written, easy to understand, and clear to read. The mathematical results look correct.
2. The newly proposed algorithm is seems somewhat novel, and it is worth further investigation whether it is indeed an improvement over existing methods.

Weaknesses:

1. The theoretical results are not strong enough. Yes, the convergence of the algorithm is established, but establishing some sort of converge rate analysis will significantly improve the strength of the paper. See 2 below:
2. The modified update rule for the dual variable (w udpate) seems to be a version of the "matrix momentum stochastic approximation" in "On Matrix Momentum Stochastic Approximation and Applications to Q-learning", Devraj et al.. It is worth investigating the connections, and more importantly, it may be worth looking at at-least the asymptotic convergence rates of the proposed GDD algorithm. Even if the scope is too large for the current paper, it may be useful for future work.
3. The numerical results are a bit concerning, and this definitely needs some changes:
  a. I am not sure why \alpha is restricted to (0,1) -- more likely than not, at-least in some experiments, large values of alpha will be beneficial. It is worth exploring values such as 10, 100, 1000, 10000. Same with \zeta values for -- it is worth exploring values large than 4 (cf. Figure 7). Specifically, It is worth trying values such as 10^-3, 10^-2, ..., 10^3. The asymptotic variance analysis suggested above will actually guide the right choice of step-size scaling.

---

> ### Author Response · Authors · 2023-12-29
>
> We appreciate this reviewer’s assessment. Below we summarize the significant modifications we have made in response to the reviewer's comments.
>
> **On the convergence rate**
>
> One concern raised is that this paper lacks of establishing some sort of convergence rate. Recognizing that a detailed analysis is beyond the scope of this paper, we have added a discussion of this issue to our Discussion section.
>
> **The ranges of $\alpha$ and $\zeta$**
>
> Concern was raised about the numerical studies regarding the ranges of $\alpha$ and $\zeta$ that were chosen.
> To clarify this, we provided additional information on why we restricted the ranges of $\alpha$ in the random walk and Boyan-chain tasks. This decision was influenced by the tendency of algorithms to diverge when $\alpha$ is relatively large.  Regarding $\zeta$, it is known that TDC approaches TD when $\zeta$ tends to 0. We have expanded our evaluation to a broader range of $\zeta$ values, spanning from $1/4^5$ to $4^4$. However, it's worth noting that when TD does not perform well, TDC with very small $\zeta$ may also exhibit poor performance. On the other hand, when $\zeta$ is large, TDC does not work well either.

---

### Review · Reviewer_xxTL · 2023-12-14

**Summary Of Contributions:**

The paper introduces a regularized mean squared projected Bellman error objective function where the regularizer penalizes large changes to the value function weights. This regularized objective function is used to derive a GTD2-like algorithm called Gradient-DD where updates to the value function weights are penalized. The paper proves the convergence of the algorithm and empirically investigates the proposed algorithm on tabular random walks, the Boyan Chain environment, and Baird’s counterexample.

**Audience:**

Yes

**Broader Impact Concerns:**

I see no broader impact concerns.

**Claims And Evidence:**

No

**Requested Changes:**

# Critical:
1. Where are the importance sampling ratios in the GTD2 and Gradient-DD derivations (sections 2, 3, and 4)? Without the importance sampling ratios, the algorithms are not adjusting for the difference in action probability between the behaviour and target policy and are instead learning about the behaviour policy.
1. It would be good to clarify what is meant by the word "stability" in the paper. E.g., "This approach is useful for the convergence guarantee, but it does not address stability issues." In off-policy policy evaluation/prediction, the word "stability" has previously been used to mean the expected update over the ergodic state distribution is a contraction by Sutton et al (2016). They also say "stability" is a prerequisite for convergence, so I'm guessing the authors mean something different than Sutton et al (2016)?
1. TDC recovers TD as the second step size $\beta$ goes to 0, so it’s very surprising that TDC performs so differently from TD. It suggests the range of values tested for $\beta$ needs to be expanded.
1. What were the best-performing values for $\alpha$ and $\beta$ for each of the algorithms in each plot? If any of them were either the maximum or minimum of the values tested, then the true best-performing parameter value could be outside the range of values tested and the range needs to be expanded. This is likely why TDC behaved so differently from TD in the experiments.
1. Section 5.1: “We tuned $\alpha$…by *minimizing the average error of the last 100 episodes*. We also compare the performance when $\alpha$ is tuned by *minimizing the average error of the last 100 episodes* and report in Fig. 5.” It seems like one of these was supposed to say something else.
1. The plots in the main paper and the corresponding plots in the appendix were substantially different and could give the reader very different impressions. For example, Gradient-DD’s performance lead in Fig 2 (main paper) vanishes in Fig 5 (appendix). Then in Fig 3 (main paper), Gradient-DD appears to be competitive with or even better than all other algorithms, but in Fig 11 (appendix) it is clearly outperformed by TD, TDRC, and even GTD2 in the left-most column. The plots where Gradient-DD didn’t perform as well should be included in the main paper, and the analysis and conclusions should be brought closer in line with the results in the appendix. For example, Conclusion 2 in Section 5.2 is not really supported by Fig 11; there is a gap between TD/TDRC and Gradient-DD, and the text should be updated to reflect this gap.
1. “Meanwhile we tune $\alpha$ for TDC in a wider region.” Could this be elaborated on? What values of $\alpha$ were tried in this experiment?
1. “minimizing the average error of the last 100 episodes” is not the right measure to support the claims made in the paper about the overall performance of the algorithms. For TD algorithms there's usually a tradeoff between early error reduction and final error level, so the parameters that result in the best final performance usually result in slower early learning and more error overall than the parameters that would minimize the total error over all episodes. However, the tradeoff is different for each algorithm, so the relative ordering of algorithms with parameters that minimize error over the last 100 episodes can be very different from the ordering of algorithms with parameters that minimize the total error over all episodes. Therefore, it’s not appropriate to draw conclusions about overall performance of the algorithms from results with parameters chosen to minimize error of the last 100 episodes. It would be better to move the results with parameters chosen to minimize final error into the appendix, move the results with parameters chosen to minimize overall error into the main paper, and update the paper to better reflect the results with parameters chosen to minimize overall error.

# Not critical:
1. “approaches *have used have used* l1 regularization”
1. Empirical study: The step size reduction schedule is something I haven’t seen before, and it prevents comparison with existing results that use constant step sizes. Constant step sizes are usually used in TD algorithms because of the non-stationarity of the TD targets when learning a value function.
1. Random Walk: Why include experiments on GTD algorithms in the tabular setting? The divergence issues that necessitate GTD don’t exist in the tabular setting and it’s not the main setting of interest for using GTD.
1. "Gradient-DD learning is an acceleration technique that employs second-order differences in successive parameter updates." It feels weird to call regularization that discourages parameter changes an acceleration technique. The conventional meaning of acceleration in optimization is using past information to make larger changes to weights, like momentum or ADAM, so using the word acceleration for a technique that causes smaller changes to weights seems confusing.

**Strengths And Weaknesses:**

# Strengths:
+ Well written and easy to follow.
+ Good explanation of the proposed algorithm.
+ Helpful connections to existing concepts such as regularization, trust region methods, and the natural gradient.

# Weaknesses:
- Empirical investigation has some issues.
- A few missing experiment details.
- Minor clarity issues.

---

> ### Author Response · Authors · 2023-12-29
>
> Thanks to this reviewer for numerous constructive comments to enhance our manuscript. We have carefully addressed each comment. Below we describe the major changes among them.
>
> **Why TDC performs so differently from TD**
>
> We have clarified the reason behind the distinct performance of TDC compared to TD. As pointed out by the reviewer, TDC tends to converge to TD when $\zeta$ approaches 0. We explain that the distinct behavior we originally observed is influenced by our choice of not taking sufficiently small $\zeta$, and we have now included the performance of TDC using a broader range of $\zeta$ values, spanning from $1/4^5$ to $4^4$.
> Additionally, we have further elaborated on the reasons behind not opting for very small $\zeta$.
>
> **The final empirical RMS error vs. the average empirical RMS error**
>
> The reviewer suggests that figures related to tuning $\alpha$ by minimizing the average error of the last 100 episodes should be placed in the appendix, while figures related to tuning $\alpha$ by minimizing the error over all episodes should be included in the main paper.
> As described in our list of major revisions above, we prefer not to make this change.
> However, we are open to switching their placement if the reviewer insists.
>
> **The results of Figures 2 and 5**
>
> We have reworked the presentation of results based on a comprehensive analysis of tuning $\alpha$, considering both the error from the final episodes and the error over all episodes.
>
> **Responses to other comments**
>
> - \#1: We have added a version of our update equations that includes the importance sampling ratios (Eqs. 7 and 12).
>
> - \#2: Regarding the concept of ``stability", our emphasis is that for off-policy learning, importance sampling methods may diverge in some cases.
> In the revised version, we have incorporated the suggestion from Reviewer 9XJi -- the motivation is centered around gradient-based methods as alternatives to semi-gradient methods, rather than solely focusing on off-policy learning.  Please review the updated introduction, particularly the first two paragraphs.
>
> - \#4: We provided additional information on why we restricted the ranges of $\alpha$ in the random walk and Boyan-chain tasks. This decision was influenced by the tendency of algorithms to diverge when $\alpha$ is relatively large.
> Regarding $\zeta$, please see our point on this in the list of major revisions above.
>
> - \#5: We have corrected the typos.
>
> - \#7: We have included a list of values that were tried in this experiment.

---

### Author Response · Authors · 2023-12-29
**Response to reviewer comments**

Thanks to the editor and to the reviewers for the helpful feedback. In our revision, we have highlighted each change in red to facilitate identification. The major weaknesses identified by the reviewers have all been addressed with substantial revisions in the revised version of our manuscript:

(1) As per the suggestion of Reviewer 9XJi, we have restructured the motivation for our work, initiating it with the general Reinforcement Learning (RL) evaluation problem instead of focusing solely on off-policy scenarios. The first two paragraphs in the introduction and the abstract have been rewritten accordingly. In this revised version, the motivation is centered around gradient-based methods as alternatives to semi-gradient methods. We've also discussed the general setting in Section 2.2 and introduced notation for off-policy learning for GTD2 in Section 2.2 and Gradient-DD in Section 3.

(2) In response to the suggestion by Reviewer 9XJi, we conducted tests to understand the implications of placing the penalty in the parameter space, rather than in the space of the value function. Figure 11 in the appendix shows that this leads to worse performance relative to Gradient-DD.

(3) As highlighted by Reviewer N9ri, a limitation of our results is the absence of a formal convergence rate analysis in our paper. Although a comprehensive analysis is beyond the scope of this paper, we have added a discussion of this issue in the Discussion section.

(4) Responding to Reviewer N9ri's concern about the possibility of using larger $\alpha$ values in our simulations, we clarified that we restricted $\alpha$ in the random walk task and the Boyan-chain task because the algorithms tend to diverge when $\alpha$ is larger than the values that we used.

(5) Reviewer xxTL expressed concern that TDC performs so differently from TD, since TDC should recover TD as $\zeta$ goes to 0.  We have expanded our evaluation to a broader range of $\zeta$ values, spanning from $1/4^5$ to $4^4$. For both the random walk and Boyan chain task, we found that the final RMS error of TDC was worse than that of TD for all values of $\zeta$ that we tried (indeed, even for smaller values that we tested but did not include in the manuscript). Hence, rather than trivially showing completely overlapping error curves for TD and TDC($\zeta=0$), we show the TDC curve with a small but nonzero $\zeta$ and provide an explanation of this in the text.

(6) Reviewer xxTL suggests that figures related to tuning $\alpha$ by minimizing the average error of the last 100 episodes should be moved from the main text to the appendix, while those tuning $\alpha$ by minimizing the overall error should be moved from the appendix to the main text. We chose to emphasize the results based on the final RMS error in order to assess the convergence performance of the algorithms. While recognizing that the reviewer's perspective on this is reasonable, we believe that our choice of how to present the results is reasonable as well. We would prefer not to swap the figures as the reviewer suggests, but we would be open to doing so before publication if the reviewer insists.
Recognizing the reviewer's point about the tradeoff between early learning speed and final performance, we have been careful not to draw unsupported conclusions in comparing the different algorithms.

As suggested by Reviewer xxTL, we have rewritten the results to include discussion on the performance both according to the final episodes (main text figures) and to all episodes (appendix figures). Specifically, in the discussions on the performance during training, we have emphasized the  distinctions in their performance evaluations under the two scenarios.

In addition to the major changes to our manuscript addressing these major issues, we have also made a number of smaller changes (also highlighted in red), following the other suggestions from each reviewer.

With these modifications, we believe that we have completely addressed the comments, concerns, and suggestions from the reviewers.

---

### Decision · Action_Editor_vXVK · 2024-02-13

**Recommendation:** Reject

**Comment:**

The reviewers have raised a number of concerns about this paper in their reviews, and further discussion has surfaced some further issues as well. A short summary is given below:

- Reviewer xxTL was seriously concerned about the value of the contributions and specifically about the rigor and the presentation style of the empirical evaluations. Their biggest concern is that the parameter range tested for some of the contending methods were cut short in an unreasonable way that favors the newly proposed algorithm. Also, for the figures included in the main text, the parameters were tuned in a questionable way that again may give the impression that GDD "outperforms" the other methods. I concur with both of these points: 1) based on the plots, there is no good reason to cut off the learning rate of GTD off at $\alpha = 1$, and 2) tuning the learning rates to optimize performance in the last 100 rounds leads to a plot where all competing methods converge more slowly to the optimum; tuning it to optimize performance over all rounds turns this picture upside down and shows that GDD is beaten by the baselines. This presentation style is arguably quite misleading, and the reviewer team did not particularly appreciate it. At the very best, the bottom line from the experiments seems to be that tuning the parameters of all these TD methods is challenging (which is hardly a new observation), and tuning the parameters of GDD is just as hard (which is not really a strong argument for the new method).

- Reviewer N9ri pointed out that there are no meaningful convergence guarantees presented in the paper. Once again, I concur with this observation, and am concerned with the validity of the main result even more than said reviewer was: one of the conditions is that $\|| \rho_n - \rho_{n+1} \||_D$ "is bounded in probability", which is not something that one can simply assume to hold. While there may be reasons to believe that this can be proved via a careful analysis, but claiming a result based on this condition and calling it a "theorem" is not acceptable in this day and age when there are so many well-established finite-time guarantees out there for TD-style methods in the literature.

- An additional point that was raised during discussion regarding the authors' claim that the proposed algorithm would be "similar" to trust-region methods like TRPO or REPS. Upon more scrutiny, this claim was deemed to be highly inaccurate and potentially misleading: while the mentioned methods enforce proximity of the subsequent policies, the newly proposed one enforces proximity of value estimates. The effects of the two regularization schemes are not comparable, and there is no reason to expect that the analysis in this paper would bring us any closer to understanding methods like TRPO / REPS or PPO as one reviewer hoped it would. If anything, the method is closer in spirit to a preconditioned version of GTD. The authors seem to be unaware of the fact that the gradient descent updates they base their method on can be derived precisely by introducing a proximal L2 regularization term in the space of parameters $w$: GD updates in general can be written as $w_{t+1} = \arg\min_w \{\langle w, \nabla f(w_t) - \frac{1}{\alpha} \||w-w_t\||_2^2\}$. Their additional regularization aims to achieve the same effect, except that it uses a weighted L2 distance in the space of value functions, which amounts to preconditioning the GD updates with the inverse covariance matrix of the features. This is not what they obtain in their algorithm because they take an "explicit" gradient descent step on the resulting objective as opposed to an "implicit" one which would give rise to the preconditioning effect I mentioned above. Interestingly, this observation *does* make the algorithm look like a "Levenberg-Marquardt-esque" trust-region method in the classical sense used in the optimization literature (see, e.g., the book of  Conn, Gould and Toint, 2000), but the authors do not hint at this connection. I think this would be an exciting direction to explore, but it would go way beyond the scope of the current submission. In fact, these same points are closely related to a paper that was mentioned by reviewer N9ri, but the authors refused to engage in that discussion about this question. Thus, it is highly unlikely that a satisfying answer to this question would be given in a new revision.

We have carefully discussed these issues with the reviewers, and eventually we have reached the consensus that this paper is not ready for publication yet. I recommend that the authors take these comments seriously when preparing a next version of the paper. In particular, they should aim to 1) present their empirical results in a more transparent way that doesn't call into question the validity of the methodology, 2) attempt to provide a more complete theoretical analysis, without excessively strong assumptions, and 3) provide a better characterization of their approach and clarify its relations with trust-region methods and preconditioned / second order stochastic approximation.

**Audience:**

The analysis of TD-like methods has always been a popular subject of reinforcement-learning research, and thus results of the kind that are presented in the paper should definitely be interesting for the TMLR readership.

**Claims And Evidence:**

While technically the claims made in the paper are all supported by some evidence, the presentation of these claims is often misleading and omit important details. See my detailed comments below.